# "No Free Lunch" in Neural Architectures? A Joint Analysis of Expressivity, Convergence, and Generalization

**Wuyang Chen**[1,*]  **Wei Huang**[2,*]  **Zhangyang Wang**[1]

[1]University of Texas at Austin
[2]RIKEN Center for Advanced Intelligence Project

**Abstract**    The prosperity of deep learning and automated machine learning (AutoML) is largely rooted in the development of novel neural networks – but what defines the "goodness" of networks in an architecture space? Test accuracy, a golden standard in AutoML, is closely related to three aspects: (1) expressivity (*how complicated* functions a network can approximate over the training data); (2) convergence (*how fast* the network can reach low training error under gradient descent); (3) generalization (whether a trained network can be generalized from the training data to unseen samples with *low test error*). However, most previous theory papers focus on fixed model structures, largely ignoring sophisticated networks used in practice. To facilitate the interpretation and understanding of the architecture design by AutoML, we target connecting a bigger picture: how does the architecture jointly impact its expressivity, convergence, and generalization? We demonstrate the "no free lunch" behavior in networks from an architecture space: given a fixed budget on the number of parameters, there does **not** exist a single architecture that is optimal in all three aspects. In other words, separately optimizing expressivity, convergence, and generalization will achieve different networks in the architecture space. Our analysis explains a wide range of observations in AutoML. Experiments on popular benchmarks confirm our analysis. Code is available at: https://github.com/chenwydj/no_free_lunch_architectures.

## 1 Introduction

Deep neural networks (DNNs) are rapidly developed in recent years. To design novel networks, Neural architecture search (**NAS**) is recently explored to remedy the human efforts and costs, benefiting automated discovery of architectures in a given search space (Zoph and Le, 2016; Brock et al., 2017; Pham et al., 2018; Liu et al., 2018a; Chen et al., 2018; Bender et al., 2018; Gong et al., 2019; Fu et al., 2020; Chen et al., 2019). To facilitate the fundamental study of automated design, many standard architecture spaces and benchmarks are also developed (Liu et al., 2018b; Ying et al., 2019; Dong and Yang, 2020). Despite the principled automation, NAS still suffers from heavy consumption of computation time and resources due to frequent training and evaluation of sampled architectures, which becomes a severe bottleneck that hinders the search efficiency.

People recently address this problem by proposing training-free NAS. Indicators like covariance of sample-wise Jacobian (Mellor et al., 2021), Neural Tangent Kernel (Chen et al., 2021), and "synflow" (Abdelfattah et al., 2021) are found to highly correlate with network's accuracy even at initialization (i.e., no gradient descent). These approaches significantly reduce search costs. However, these works mainly leverage theoretical properties of the general deep neural networks in experiments, but barely characterize the inductive bias of these indicators on network architectures.

Meanwhile, many deep learning theory papers try to understand deep networks. A typical pipeline of learning involves three components: 1) data (or task), 2) network training (with gradient descent), and 3) inference (on unseen data). First, given the training data, the network needs to be

---

*Equal Contribution

highly expressive to approximate the target function with a low training error. Second, the network should converge fast in an affordable training time. Third, the network should not simply memorize the training samples, but needs to be generalizable to unseen data during inference. Being deficient in any aspect would lead to the failure of utilizing deep networks (low accuracy, slow convergence, overfitting, etc.). However, most theory papers focus on analyzing a fixed model structure, largely ignoring **sophisticated networks** used in practice, especially for NAS applications and standard architecture spaces and benchmarks we mentioned above. This leads to a concrete question:

*Q1: What are the inductive biases of expressivity, convergence, and generalization on networks in an architecture space? Do they prefer wide or deep network topologies?*

This motivates us to jointly analyze how these three aspects change accordingly when we design network topologies in an architecture space. In addition, most previous theory works only study one or two aspects at the same time, which may not reveal the global picture of a network's property. Imagining we are searching and optimizing the network architecture for its expressivity, convergence, and generalization together at the same time, a further question is also unclear:

*Q2: When we design networks in an architecture space, can we achieve the best expressivity, convergence, generalization at the same time? In other words, can we find an architecture that will "win" all three aspects? Or do we have to sacrifice one or two of them to compensate the others?*

Inspired by recent complicated networks designed in NAS, and to facilitate the interpretation and understanding of architectures design by AutoML, we rigorously study the impact of a network's topology on its expressivity, convergence, generalization in an architecture space. Network architecture can be viewed as a directed acyclic computational graph (DAG), where feature maps are represented as nodes and operations in different layers are directed edges linking features.

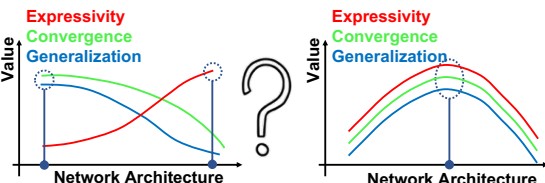

Figure 1: Is there a network in an architecture space that can achieve the best expressivity, convergence, generalization at the same time?

We discover the **"no free lunch"** behavior: given a fixed budget on the number of parameters, there does **not** exist such an architecture that can maximize all three aspects. We first abstract an architecture's graph structure into its topological width and depth. By analyzing the input-output Jacobian, NNGP (Neural Network Gaussian Process), and NTK (Neural Tangent Kernel) of ReLU networks with a large channel width, we can characterize the dependence of manifold complexity, convergence rate, and generalization gap on the network's graph topology. After finding corresponding architectures that maximize three aspects, we show that both convergence and generalization have a bias toward networks with wide and shallow graph topologies, but the expressivity favors deep and narrow ones. Our analysis can explain a wide range of observations in AutoML and NAS. Experiments on popular vision benchmarks confirm our theoretical analysis. Our contributions are summarized below:

- We theoretically analyze the dependence of a deep network's manifold complexity, convergence rate, and generalization gap on its graph topology.
- We discover the **"no free lunch"** behavior: given a fixed budget on the number of parameters predefined in an architecture space, no such a network can achieve optimal expressivity, convergence, and generalization at the same time.
- Our analysis can explain a wide range of observations in AutoML and NAS practices. Experiments on popular datasets confirm our theoretical analysis.

## 2 Related Works

### 2.1 Theory-guided Automated Design of Neural Architectures

**Neural architecture search (NAS)** is proposed to accelerate the principled and automated discovery of high-performance networks (Pham et al., 2018; Liu et al., 2018b; Dong and Yang, 2019; Real et al.,

2019; Tan et al., 2020). However, most works suffer from heavy search cost. Therefore, recent research on NAS has shifted its focus towards reduced training or even training-free methods. The aim is to connect theoretical analysis of deep learning to guide the development of innovative network architectures. The key idea is to identify theoretical indicators that are highly correlated with the network's training or testing performance. Mellor et al. (2020) introduced a training-free NAS approach that uses sample-wise activation patterns to rank architectures empirically. Park et al. (2020) used the network's NNGP features to estimate its predictions. Various training-free indicators were evaluated in (Abdelfattah et al., 2021), and the "synflow" measure was adopted in (Tanaka et al., 2020) as the primary ranking metric. Chen et al. (2021) incorporated two metrics inspired by theory, and used supernet pruning as the search method. Li et al. (2023) further discovered the norm of network's gradient over the gradient variance as an accurate proxy indicator.

Despite the inspiring result, these works mainly leveraged theoretical properties of the general deep neural networks in experiments, but barely characterize the inductive bias of these indicators on network architectures. In our work, we try to connect network topologies (defined in a typical graph-based architecture space) with decoupled properties of networks (discussed below).

## 2.2 Expressivity, Convergence, and Generalization of Network Architectures

Many works try to theoretically characterize the deep network's properties, including expressivity (Poole et al., 2016; Hanin and Rolnick, 2019a,b; Hanin et al., 2021; Fawzi et al., 2018), convergence (Allen-Zhu et al., 2019b; Du et al., 2019; Lu et al., 2020; Zou et al., 2020a; Zhou et al., 2020; Zou et al., 2020b), and generalization gap (Neyshabur et al., 2015; Bartlett et al., 2017; Arora et al., 2018; Wei et al., 2019; Xiao et al., 2019; Cao and Gu, 2019; Allen-Zhu et al., 2019a; Zhang et al., 2021).

- **Expressivity**. Classic works focus on proving the existence of networks with low approximation error, demonstrating the benefit of network depths (Telgarsky, 2016; Eldan and Shamir, 2016; Rolnick and Tegmark, 2017; Park et al., 2021). Layer-wise recursion of the network's length distortion and extrinsic curvature in Riemannian geometry is given (Poole et al., 2016). The network's depth, spectrum, linear regions, persistent homology are also studied (Bianchini and Scarselli, 2014; Lu et al., 2017; Rieck et al., 2018; Rahaman et al., 2019; Hanin and Rolnick, 2019a).

- **Convergence**. The rate to converge to the global minima of MLP and ResNet is given (Du et al., 2019), and skip-connection can improve (reduce) the requirement on the channel number to be polynomial of the network depth, without requiring the network to be exponentially wide. In (Bhardwaj et al., 2021), the network topology, or specifically, the number of skip-connections, is found to improve the network's training convergence and layerwise dynamical isometry, from a network science perspective. In addition, the variance of the network's output and gradient are proved to scale as the depth-to-width ratio, i.e., the effective depth, instead of the absolute network depth (Hanin, 2022). The convergence rates of stochastic neural networks with different hyper-parameters have been studied in (Huang et al., 2023). The convergence rates of different network architecture topologies are recently compared (Chen et al., 2022).

- **Generalization**. The gap between the training and testing accuracy is empirically found relevant to network topology. Some common connectivity patterns are discovered by neural architecture search and can contribute to fast convergence, high test accuracy, and smooth loss landscapes (Shu et al., 2019). Structures of networks are represented into graphs, and then discovered that networks of specific graph topology can achieve strong test accuracy (You et al., 2020).

In contrast, we comprehensively unify the analysis of expressivity, convergence, and generalization, and demonstrate the "no free lunch" behavior of architectures on these three aspects.

## 3 "No Free Lunch"[*] in Network Architectures

In this section, we introduce the high-level idea of the "no free lunch" behavior of networks from an architecture space. We first define the graph topology of networks (Section 3.1), and then give the high-level results on how architecture impacts its expressivity, convergence, generalization (Section 3.2). We defer detailed formal statements in (Section 4) for paper organization purpose.

### 3.1 Graph Topology of Neural Architectures

**Graph-based architectures spaces**. The computational graph of a neural network can be viewed as a directed acyclic graph (DAG). Nodes are inputs or features, and edges are layers (operations). Features from multiple edges coming into one node will be summed up. The graph's connectivity pattern is allowed to be arbitrary: any two nodes can be connected by an edge. Recent cell-based architecture spaces (Liu et al., 2018b; Ying et al., 2019; Dong and Yang, 2020) also leverage this graph formulation of networks. In our work, we inherit and simplify these architecture spaces, and consider three types of edges (Figure 2): linear transformation with ReLU activation, skip connection, and zero.

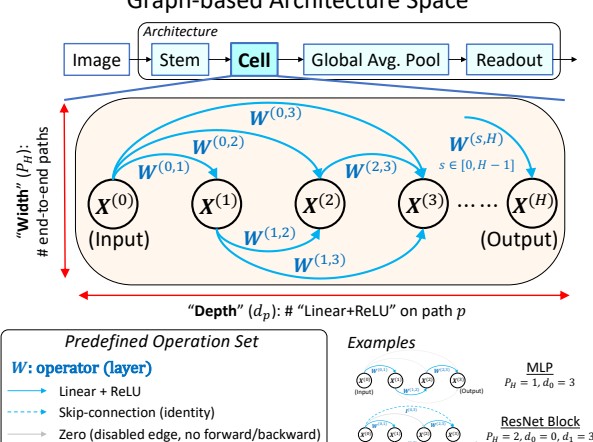

Figure 2: Graph formulation of architectures. $X$: input/feature/output (node). $W$: layers (edge). **Top**: shared macro skeleton across architectures. **Middle**: graph formulation of cell in architectures. **Bottom left**: each edge is an operation from a predefined set. **Bottom right**: example architectures with their topological depths and widths.

**Graph topology**. In our architecture space, an end-to-end path is defined as a finite sequence of edges that joins the input ($X^{(0)}$) and the output node ($X^{(H)}$). We denote the number of end-to-end unique paths as $P_H$, and the number of linear transformation operations on the $p$-th path ($p \in [1, P_H]$) as $d_p \in [0, H]$. Intuitively, $P_H$ stands for the "**topological width**" of a graph, and $d_p$ represents the "**topological depth**" of a graph.

### 3.2 Architecture Biases of Expressivity, Convergence, and Generalization

In automated machine learning (AutoML) and neural architecture search (NAS), what defines and controls the "goodness" of architectures? In fact, test accuracy, a golden standard by AutoML practitioners, can be disentangled and is closely related to three key properties: functional complexity a network can approximate ("expressivity"), training speed under gradient descent ("convergence"), and performance gap between training and unseen data ("generalization"). Here, we introduce our high-level results on their biases on the network's architecture and highlight impacts on broad AutoML applications. For detailed formal statements, please see Section 4.

#### 3.2.1 Expressivity. 
We characterize the expressivity of a neural network using concepts in Riemannian geometry (Lee, 2006). Consider the mapping from each point in the input space to the network's output (the manifold), the curvature of the manifold indicates how quickly its tangent vector rotates as one moves across the input space. Intuitively, if a network has highly curved output manifolds, it may have a higher capacity to learn complex functions and decision boundaries (Poole et al., 2016).

We will show that given a space of architectures, with standard He normal initialized weights (He et al., 2015), the architecture of small $P$ and large $d_p$ will have large curvature (for details, see Theorem 4.1 and Corollary 4.2).

---

[*]Originally proposed (Wolpert, 1996) to explain the equivalence of algorithms over learning problems.

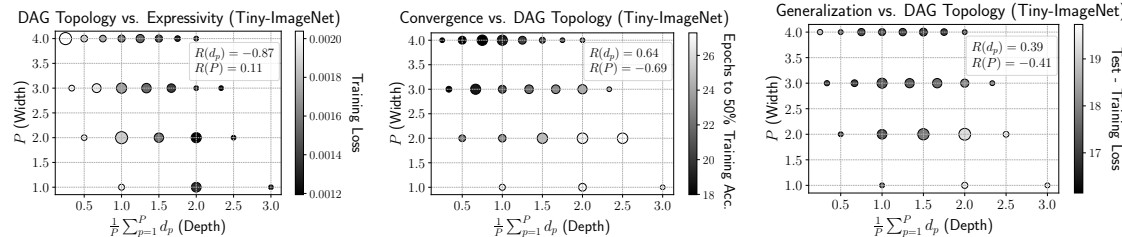

Figure 3: Given a fixed budget of the number of parameters, networks of deep and narrow topology have higher **expressivity** (converged training loss) (left), while wide and shallow ones show faster **convergence** (number of epochs to reach 50% training accuracy) (middle) and smaller **generalization gap** (gap between converged test and training loss) (right). All 729 networks (when $H = 3$) are trained on Tiny-ImageNet. Small values (dark circles) the better. Kendall-tau correlations ("$R$") are reported in legends. Radiuses indicate standard deviations over networks of the same graph topology ($P$ and $\frac{1}{P}\sum_{p=1}^{P} d_p$).

### 3.2.2 Convergence Rate.
A network with a fast convergence rate will reach a low training error in fewer training iterations. Given a space of architectures training with gradient descent, the architecture of large $P$ and small $d_p$ will have fast convergence rate (for details, see Theorem 4.4).

### 3.2.3 Generalization.
Generalization is defined as the **gap** between errors on the training set and testing set, i.e., how much a network can learn generalizable predictions on unseen data. A network of poor generalization (large gap) will just memorize the training data and lead to overfitting.

Given a space of architectures training with gradient descent, the architecture of large $P$ and small $d_p$ will have small generalization gap (for details, see Theorem 4.5 and 4.6).

### 3.2.4 Experiments[†].
To verify our above analysis on the network's expressivity, convergence, and generalization, we choose an architecture space of $H = 3$ (which follows Dong and Yang (2020)), train all 729 networks[‡] and keep the training dynamics of all networks on Tiny Imagenet (Tin, 2015). Results in Figure 3 confirm our analysis. On all three plots, the darker circles are better: we want lower training loss, fewer epochs to reach a fair amount of accuracy, and we want smaller generalization gaps. We can observe that, for the expressivity, all dark circles locate on the bottom right, meaning that they are deep and narrow graphs. However, for both trainability and generalization, darker circles are on the top left, indicating the wide shallow graphs. For results on CIFAR-10 and CIFAR-100 please refer to our Appendix C.1 in the supplement.

### 3.2.5 "No Free Lunch".
The above results conclude: maximizing the convergence rate and minimizing the generalization gap will lead to networks of wide and shallow graph topology in an architecture space. In contrast, maximizing the network's expressivity (manifold curvature) will reach narrow and deep topologies. This reveals a **"no free lunch"** behavior in the network architecture: it **cannot** achieve the best in all three aspects, but has to maintain a balance.

**Implications**. The "no free lunch" behavior can explain important AutoML and NAS applications:

- Architecture bias in differentiable NAS. Differentiable methods are found to have an intrinsic bias to choose more skip-connections than parameterized layers during the neural architecture search (Zela et al., 2019; Zhou et al., 2020; Ye et al., 2022; Chu et al., 2020b,a). This bias leads to the collapse issue, an undesirable phenomenon associated with DARTS (Liu et al., 2018b), as noted by (Liang et al., 2019): excessive skip-connections in chosen architectures lead to shallower networks with fewer learnable parameters compared to deeper ones, ultimately resulting in

---

[†]See Appendix A for experiment settings.

[‡]For $H = 3$, there are in total 6 possible edges in the cell 2, with each edge having 3 possible layer types (see Eq. 2: "Linear + ReLU", "Skip-connection", and "Zero"), thus $3^6 = 729$.

reduced expressive power. This observation is corroborated by common connectivity patterns identified in (Shu et al., 2019): "architectures generated by popular NAS algorithms tend to have the widest and shallowest cells among all candidate cells in the same search space." Based on our analysis, this is mainly because architecture configurations are optimized concurrently with shared weights in the bi-level optimization in DARTS. That means architectures and operations are selected based on network parameters that are not yet fully trained, which reflect more on their **convergence** property instead of expressivity or generalization! Therefore, the differentiable search tends to favor networks that can minimize training loss as quickly as possible. The skip-connection can increase a network's topological width (more paths) and reduce its depths, thus being more favorable to the convergence of the differentiable search.

- Good networks are of balanced depth/width. From an architecture space, given a fixed budget of the number of parameters, networks of moderate depth and width (instead of being too wide or too deep) show better performance (Figure 5 in (Chen et al., 2022)). This empirical observation can be explained by our theoretical justification: because of a *comprehensive effect* of expressivity, convergence, and generalization (which all contribute to the final performance), their intrinsic trade-offs require the architecture to balance all three aspects. Since expressivity, convergence, and generalization have different architecture biases, this further requires the network to balance its depth and width. This intrinsic trade-off on depth/width can further facilitate AutoML (Section 5.3 in (Chen et al., 2022)): simply pursuing a balanced network topology can speed up the training-free neural architecture search.

- Neural Scaling Law (Kaplan et al., 2020). When people try to scale up large models, we cannot just simply add more layers or more channel widths, but we have to do both, to balance the network's depth/width and thus three properties. This can be attributed to the aforementioned reason, that expressivity, convergence, and generalization all contribute to the final performance but have different biases on architectures. As claimed in (Kaplan et al., 2020), "models with fewer than 2 layers or with extreme depth-to-width ratios deviate significantly from the trend." This observation is supported by Figure 6 in their study, which demonstrates that when the number of parameters is held constant, networks with too few layers underperform, while overly deep networks converge to a singular loss curve. In a similar vein, Figure S5 in (Bahri et al., 2021) reveals that when the width factor is fixed at 10, networks with moderate depths (16, 28) exhibit lower loss than those with extreme depths (10, 40).

*Remark* 3.1. We shall emphasize that we compare different networks in a complete architecture space: given a fixed number of nodes (feature maps), the maximally possible number of edges (layers, or neurons) is also fixed. That means, a network can either allocate its neurons to its width or its depth, but it cannot be both the widest and deepest in the space. The "no free lunch" behavior states that given the same number of parameters (for a fair comparison), expressivity and trainability/generalization pursue different choices of topological depth and width.

## 4 Formal Results

In this section, we provide formal definitions and statements of our results on expressivity, convergence, and generalization to explain our core result in Section 3. Full proofs are given in the Appendix D, E, and F in supplement. Note that, although being at the network's initialization, our analysis can reflect the inductive bias of architectures to the expressivity, convergence, and generalization. Moreover, we also provide experiments to verify our theoretical analysis (Figure 4, and more in the supplement).

### 4.1 Problem Setup and Architectures Notations (for the Graph Topology in Section 3.1)

We consider the computational graph of a network illustrated in Figure 2. $X^{(0)}$ is the input node, $X^{(H)}$ is the output node, and $X^{(1)}, \cdots, X^{(H-1)}$ are intermediate nodes (feature maps). $W$ is the

layer operation (edge). The forward process of the network in Figure 2 can be formulated as below.

$$X^{(t)} = \sum_{s=0}^{t-1} \rho(W^{(s,t)} X^{(s)}) \qquad (t \in [1, H]) \tag{1}$$

$X \in \mathbb{R}^{m \times 1}$, where $m$ is the absolute width of a layer. In our analysis, each layer (edge) can choose from three operations: 1) linear transformation followed by a ReLU activation, 2) a skip-connection (the identity mapping), 3) a zero mapping (broken edge, no forward and backward allowed):

$$W \begin{cases} = \mathbf{0} & \text{zero} \\ = I^{m \times m} & \text{skip-connection} \\ \sim \mathcal{N}(\mathbf{0}, I^{m \times m}) & \text{linear transformation} \end{cases}, \quad \rho(x) = \begin{cases} 0 & \text{zero} \\ x & \text{skip-connection} \\ \sqrt{\frac{c_\sigma}{m}} \sigma(x), & \text{linear transformation} \end{cases} \tag{2}$$

$\mathcal{N}$ stands for the Gaussian distribution for weight initialization, $\sigma$ represents the ReLU activation, and we set $c_\sigma = 2$ (Hayou et al., 2019).

## 4.2 Expressivity Analysis of Architectures (for Section 3.2.1)

We study the functional complexity for deep networks. Our goal is to compare the expressivity of different networks and establish links to their graph topologies. Following (Poole et al., 2016), we consider a simple circle input $X^{(0)}(\theta) = \sqrt{N_0} [\mathbf{u}_0 \cos(\theta) + \mathbf{u}_1 \sin(\theta)]$, where $\theta \in [0, 2\pi)$, $\mathbf{u}_0$ and $\mathbf{u}_1$ form an orthonormal basis for a 2-dimensional subspace of the input space $\mathbb{R}^{N_0}$ (e.g. $N_0 = 3 \times 32 \times 32$ for images in CIFAR-10 dataset).

We first demonstrate how the network's graph topology impacts its norm of input-output Jacobian.

**Theorem 4.1** (Jacobian in Architectures). *For ReLU networks in our architecture space (Figure 2) of nodes $0, 1, \cdots, H$. The total number of end-to-end paths is $P_H$, and the depth of each path is $d_p$ ($p = 1, \cdots, P_H$). Weights are initialized by the standard He normal initialization (He et al., 2015). The expectation (over the weight distribution) of the Jacobian's norm of this network is:*

$$\int_0^{2\pi} \mathbb{E}\left[\|\mathbf{J}(\theta)\|\right] d\theta = C \cdot \sum_{p=1}^{P_H} \exp\left[-\frac{5}{8}\frac{d_p}{m} + O\left(\frac{d_p}{m^2}\right)\right], \tag{3}$$

*where $C = \dfrac{\Gamma\left(\frac{m+1}{2}\right)}{\Gamma\left(\frac{m}{2}\right)\left(\frac{m}{2}\right)^{1/2}}$, $m$ is the hidden layer width (Eq. 2), and $\Gamma(\cdot)$ denotes the Gamma function.*

An important reason why we choose manifold curvature and Jacobian to indicate the network's expressivity is to seek an **average**-case analysis (Theorem 4.1). Classic approximation theories (which prove the existence of networks that can approximate certain functions with low errors) focus on the best-case analysis (i.e. the existence of a certain network that satisfies low approximation error). The best-case analysis considers the maximum complexity of functions that may be expressed by the network by varying its parameters. In contrast, the average-case analysis considers the typical complexity of the network with a given distribution of parameters. It is now increasingly recognized that average-case analysis better reflects a network's inductive biases (Hanin and Rolnick, 2019a; Hanin et al., 2021).

Finally, to characterize the expressivity of networks via manifold curvature, we show that a high curvature requires a small norm of Jacobian in ReLU MLP networks.

**Corollary 4.2** (Curvature and Jacobian). *For a ReLU network, given a unit circle input (Poole et al., 2016), we have its curvature as the reciprocal of the norm of the input-output Jacobian:*

$$\kappa(\theta) = \|\mathbf{J}(\theta)\|^{-1} \tag{4}$$

Our goal is to maximize $\kappa(\theta)$ (i.e. to minimize $\|\mathbf{J}(\theta)\|$) via morphing the network's graph topology. Theorem 4.1 indicates that networks of more short paths (i.e. wide and shallow graphs) will more likely have a larger norm of input-output Jacobian. This is because the left hand side of Eq. 3 can be enlarged by increasing $P_H$ (wider) and reducing $d_p$ (depth). And based on Corollary 4.2, we can see that deep and narrow networks will have higher curvature, which is aligned with their low training error in Figure 3. Besides, our analysis can be further confirmed by:

- Our analysis can find the network of the highest expressivity: in our architecture space in Figure 2, when $H = 3$, the largest depth is 3 with a small width of 1. This architecture indeed has the highest manifold curvature in our experiment, and also shows the lowest training loss at converge.

- We also visualize $\kappa(\theta)$ vs. the graph topology. In Figure 4 left, we can indeed see that deeper and narrower networks have higher $\kappa(\theta)$. This trend is aligned with Figure 3 left.

It is worth noting the core difference between the manifold curvature $\kappa$ and Hessian: the curvature characterizes the sensitivity of the network's output to its input, whereas the Hessian characterizes the sensitivity of the network's output to its parameters. Since the expressivity indicates whether a network can learn complicated mappings from its input space to output space, the manifold curvature is a more precise characterization than Hessian.

### 4.3 Convergence Analysis of Architectures (for Section 3.2.2)

We formally link the network's convergence rate with its graph topology. We first follow the bound of convergence rate by a network's least eigenvalue of it NNGP kernel (Chen et al., 2022).

**Theorem 4.3** (Linear Convergence of Architectures (Chen et al., 2022)). *Consider an architecture of $H$ nodes and $P_H$ end-to-end paths. At $k$-th gradient descent step on $N$ training samples, with MSE loss $L(k) = \frac{1}{2}\|\mathbf{y} - \mathbf{X}^{(H)}(k)\|_2^2$, suppose the learning rate $\eta = O\left(\frac{\lambda_{\min}(\mathbf{K}^{(H)})}{(NP_H)^2}2^{O(H)}\right)$ and the number of neurons per layer $m = \Omega\left(\max\left\{\frac{(NP_H)^4}{\lambda_{\min}^4(\mathbf{K}^{(H)})}, \frac{NHP_H}{\delta}, \frac{(NP_H)^2\log(\frac{HN}{\delta})2^{O(H)}}{\lambda_{\min}^2(\mathbf{K}^{(H)})}\right\}\right)$, we have*

$$\|\mathbf{y} - \mathbf{X}^{(H)}(k)\|_2^2 \leq \left(1 - \frac{\eta\lambda_{\min}(\mathbf{K}^{(H)})}{2}\right)^k \|\mathbf{y} - \mathbf{X}^{(H)}(0)\|_2^2. \tag{5}$$

$\mathbf{K}_{ij}^{(H)} = \langle \mathbf{X}_i^{(H)}, \mathbf{X}_j^{(H)} \rangle$ *is the network's NNGP kernel at node $H$ ($i, j \in [1, N]$), whose expectation is taken over the network's random initializations. $P_H$ is number of end-to-end paths from $\mathbf{X}^{(0)}$ to $\mathbf{X}^{(H)}$.*

This means larger $\lambda_{\min}(\mathbf{K}^{(H)})$ indicates faster convergence. Next, we give the theorem that bound the NNGP's least eigenvalue by the network's graph topology.

**Theorem 4.4** ($\lambda_{\min}(\mathbf{K}^{(H)})$ of Architectures). *For ReLU networks in our architecture space (Figure 2) of nodes $0, 1, \cdots, H$. The total number of end-to-end paths is $P_H$, and the depth of each path is $d_p$ ($p = 1, \cdots, P_H$). The least eigenvalue of NNGP kernel of this network is:*

$$\lambda_{\min}(\mathbf{K}^{(H)}) \leq \min_{i \neq j}\left[P_H - \sum_{p=1}^{P_H} f^{d_p}(\mathbf{K}_{ij}^{(0)})\right] \quad i, j \in [1, N]. \tag{6}$$

In Eq. 33, $f$ is a function (defined in Appendix F.4) that characterizes how the NNGP kernel $\mathbf{K}$ propagates through layers in a ReLU network. $f(\mathbf{K}_{ij}) > \mathbf{K}_{ij}$ and $f(\mathbf{K}_{ij}) \in [0, 1)$. We also define $d_p$-power composition of a function as $f^{d_p} = \overbrace{f \circ f \circ \cdots \circ f}^{d_p}(\cdot)$. We target morphing the network's architecture to maximize $\lambda_{\min}(\mathbf{K}^{(H)})$, and we have the following steps:

1. Whenever we add one more path, $P_H$ will increase by 1, and $\sum_{p=1}^{P_H} f^{d_p}(\mathbf{K}_{ij}^{(0)})$ will increase one more term. For a single path, $f^{d_p}(\mathbf{K}_{ij}^{(0)}) \in [0, 1)$. We can thus guarantee to improve $\lambda_{\min}(\mathbf{K}^{(H)})$ by a positive margin by adding more paths. Therefore, we should first maximize $P_H$.

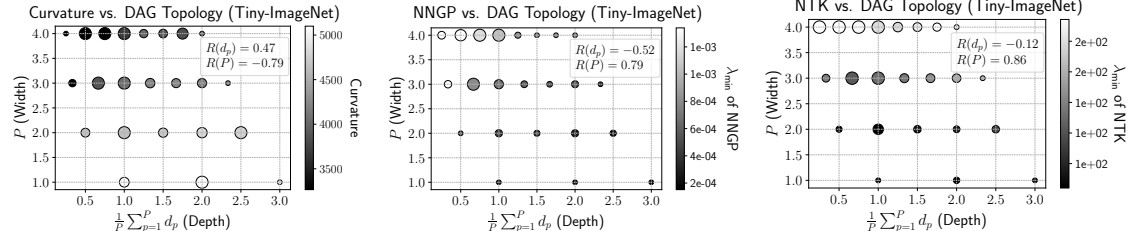

Figure 4: Given a fixed budget of the number of parameters in an architecture space, networks of deep and narrow topologies have higher **expressivity** (manifold curvature) (left), while wide and shallow ones have both large **convergence rate** ($\lambda_{\min}$ of NNGP) (middle) and smaller **generalization gap** ($\lambda_{\min}$ of NTK) (right). Larger the values (white circles) the better in all three plots. Kendall-tau correlations ("$R$") are reported in legends. Radiuses indicate standard deviations over networks of the same graph topology ($P$ and $\frac{1}{P}\sum_{p=1}^{P} d_p$). Curvature and $\lambda_{\min}$ of NNGP/NTK are averaged over three random runs.

2. After fixing $P_H$ to be the maximal number of paths in a graph, we should minimize $\sum_{p=1}^{P} f^{d_p}(K_{ij}^{(0)})$, i.e., to put as fewer as number of linear transformations on the edges[§].

Only having an upper bound may not be enough to confirm the dependence of $\lambda_{\min}(K^{(H)})$ on the graph topology. To demonstrate that this upper bound is meaningful, we also visualize $\lambda_{\min}(K^{(H)})$ vs. graph topology in experiments. In Figure 4 middle, we can see that wider and shallower networks have higher $\lambda_{\min}(K^{(H)})$. The theoretical and experimental analysis tells us that networks that are wider (larger $P_H$) and shallower (smaller $\sum_{p=1}^{P_H} f^{d_p}(K_{ij}^{(0)})$) will more likely to converge earlier, which is aligned with their faster convergence in Figure 3 middle.

Besides, our analysis can find the network of the best trainability: in our architecture space in Figure 2, when $H = 3$, the largest number of unique paths is 4, with the smallest averaged depths as 0.25. This architecture indeed has the fastest convergence speed in our experiment.

## 4.4 Generalization Analysis of Architectures (for Section 3.2.3)

In this section, inspired by recent works (Arora et al., 2019b; Cao and Gu, 2019), we give the generalization bound of architectures via neural tangent kernel (NTK) (Jacot et al., 2018) and Rademacher complexity in the over-parameterization regime. We then analyze the architecture's impact on this generalization bound.

**Theorem 4.5** (Generalization of Architectures). *Suppose dataset $S = \{(\boldsymbol{x}_i, y_i)\}_{i=1}^{N}$ are i.i.d. samples from a non-degenerate distribution $\mathcal{D}(\boldsymbol{x}, y)$, and $m \geq \text{poly}(N, \sum_{p=1}^{P_H} d_p, \lambda_{\min}^{-1}(\boldsymbol{G}^{(H)}), \delta^{-1})$. Consider any loss function $\ell : \mathbb{R} \times \mathbb{R} \to [0,1]$ that is 1-Lipschitz, then with probability at least $1 - \delta$ over the random initialization, the network trained by gradient descent for $K \geq \Omega(\frac{1}{\eta \lambda_{\min}(\boldsymbol{G}^{(H)})} \log \frac{N}{\delta})$ iterations has population risk $L_{\mathcal{D}} = \mathbb{E}_{(\boldsymbol{x}, y) \sim \mathcal{D}(\boldsymbol{x}, y)}[\ell(f(\boldsymbol{x}; K)), y)]$ that is bounded as follows:*

$$L_{\mathcal{D}} \leq \widetilde{O}\left(\left(\sum_{p=1}^{P_H} d_p\right) \cdot \sqrt{\frac{\boldsymbol{y}^{\top}(\boldsymbol{G}^{(H)})^{-1}(X, X)\boldsymbol{y}}{N}}\right) + O\left(\sqrt{\frac{\log(1/\delta)}{N}}\right) \tag{7}$$

*where $\boldsymbol{G}^{(H)} = \langle \frac{\partial \boldsymbol{X}^{(H)}}{\partial \boldsymbol{W}}, \frac{\partial \boldsymbol{X}^{(H)}}{\partial \boldsymbol{W}} \rangle$ is the NTK of the network, and $\boldsymbol{W}$ is the collection of all weights. We use $\widetilde{O}(\cdot)$ to hide the logarithmic factors in $O(\cdot)$.*

We leave the proof of the above theorem in the Appendix F.1. Since the leading term of the generalization bound is $(\sum_{p=1}^{P_H} d_p) \cdot \sqrt{\frac{\boldsymbol{y}^{\top}(\boldsymbol{G}^{(H)})^{-1}(X, X)\boldsymbol{y}}{N}}$ and all networks in our architecture space share the same data and labels, we compare the generalization bound for different networks based on the following inequality:

---

[§]We need to make sure $\sum_{p=1}^{P} d_p > 0$, since a network of no parameterized layers will not learn anything.

$$\left(\sum_{p=1}^{P_H} d_p\right) \cdot \sqrt{\frac{\boldsymbol{y}^\top (G^{(H)})^{-1}(X,X)\boldsymbol{y}}{N}} \le \sum_{p=1}^{P_H} d_p \cdot \frac{1}{\sqrt{\lambda_{\min}(G^{(H)})}} \tag{8}$$

We can recursively compute the NTK from the NNGP (Jacot et al., 2018; Arora et al., 2019c):

$$G^{(h)} = K^{(h)} + \dot{K}^{(h)} G^{(h-1)}, \quad G^{(0)} = K^{(0)} \tag{9}$$

where $\dot{K}^{(h)} = \langle \dot{\rho}(W^{(h-1,h)}X^{(h-1)}), \dot{\rho}(W^{(h-1,h)}X^{(h-1)})\rangle$ and $h \in [1, H]$. Finally, we give the theorem that bounds the NTK's least eigenvalue by the network's graph topology.

**Theorem 4.6** ($\lambda_{\min}(G^{(H)})$ of Architectures). *For ReLU networks in our architecture space (Figure 2) of nodes $0, 1, \cdots, H$. The total number of end-to-end paths is $P_H$, and the depth of each path is $d_p$ ($p = 1, \cdots, P_H$). The least eigenvalue of NTK kernel of this network is:*

$$\lambda_{\min}(G^{(H)}) \le \min_{i \ne j} \left[ P_H - \sum_{p=1}^{P_H} f^{d_p}(K_{ij}^{(0)}) + \sum_{p=1}^{P_H} d_p - \sum_{p=1}^{P_H} \sum_{e=1}^{d_p} f^e(K_{ij}^{(0)}) \prod_{k=1}^{e} \dot{f}^k(K_{ij}^{(0)}) \right] \quad i, j \in [1, N]. \tag{10}$$

In Eq. 55, $\dot{f}$ is a function (defined in Appendix F.2) that characterizes how fast the NNGP kernel $K$ propagates through layers in a ReLU network, and $\dot{f}(K_{ij}) \in [0, 1)$ given $K_{ij} \in [0, 1)$ (Hayou et al., 2019). Note that $\dot{f}^k(K_{ij}^{(0)}) = \partial f^k(K_{ij}^{(0)})/\partial f^{k-1}(K_{ij}^{(0)})$. We then target on how to morph the network's architecture to minimize the leading term in the generalization bound in Eq. 8. Basically, we need to minimize both $\sum_{p=1}^{P_H} d_p$ and $\sqrt{\frac{\boldsymbol{y}^\top (G^{(H)})^{-1}(X,X)\boldsymbol{y}}{N}}$ (i.e., to maximize $\lambda_{\min}(G^{(H)})$):

1. To minimize $\sum_{p=1}^{P_H} d_p$ in Eq. 8: we prefer shallower graph structures when $P_H$ is fixed.

2. To maximize the upper bound of $\lambda_{\min}(G^{(H)})$, we consider two groups in Eq. 55:

   - $P_H - \sum_{p=1}^{P_H} f^{d_p}(K_{ij}^{(0)})$: this group is the same as NNGP in Eq. 33, which favors wide and shallow graph structures.

   - $\sum_{p=1}^{P_H} d_p - \sum_{p=1}^{P_H} \sum_{e=1}^{d_p} f^e(K_{ij}^{(0)}) \prod_{k=1}^{e} \dot{f}^k(K_{ij}^{(0)})$: these two are extra terms introduced by NTK. As $d_p \to \infty$, we have $f^{d_p}(K_{ij}^{(0)}) \to 1$ and $\dot{f}^k(K_{ij}^{(0)}) \to 1$. Therefore, as $d_p$ grows, two terms in this group compete with each other, and will be canceled in a large limit of $d_p$.

Only having an upper bound may not be enough to confirm the dependence of $\lambda_{\min}(G^{(H)})$ on the graph topology. To demonstrate that this upper bound is meaningful, we also visualize $\lambda_{\min}(G^{(H)})$ vs. graph topology in experiments. In Figure 4 right, we can see that wider and shallower networks have higher $\lambda_{\min}(G^{(H)})$. The theoretical and experimental analysis tells us that wider and shallower networks (larger $P$, smaller $\sum_{p=1}^{P} d_p$) will more likely to have lower generalization gap, which is aligned with their low "test − training error" in Figure 3 right.

## 5 Conclusion and Discussions

To facilitate the explanation of the architecture bias in AutoML and NAS applications, in this work, we jointly analyze a network's expressivity, trainability, and generalization, and how they are influenced by the architecture's graph topology. Given a fixed budget of the number of parameters, we show that the expressivity favors networks of deep and narrow graph topologies, whereas both the trainability and generalization prefer wide and shallow ones. We for the first time discover that these inductive biases lead to a "no free lunch" behavior in deep network architectures: we cannot achieve the best over all three aspects in one network.

We identify three limitations in our current methods and results. First, different networks may not necessarily share the same optimal learning rate, and the comparison under the same training protocol may be unfair. Second, finding a "golden standard" for characterizing the expressivity of a neural network is still challenging. Finally, our analysis still focuses on the network's initialization stage, and characterizing the training dynamics is still meaningful but challenging.

## 6 Acknowledgement

Z. Wang is supported by NSF Scale-MoDL (award number: 2133861).

## 7 Broader Impact Statement

Our work has a significant impact on the research community of automated machine learning and neural architecture search. Overall, our work sheds light on the limitations and challenges of optimizing neural networks in an architecture space. We challenge the prevailing assumption that there exists a single optimal network architecture and instead highlight the importance of balancing these three aspects when optimizing neural networks.

Our research is crucial for improving the performance of automated machine learning and neural architecture search algorithms, as we provide a theoretical foundation for designing better architectures. Additionally, our work has the potential to stimulate further research and innovation in AutoML and NAS, as researchers seek to optimize their algorithms by balancing expressivity, convergence, and generalization. Moreover, the impact of our work extends beyond AutoML and NAS to the broader field of deep learning, as it advances our understanding of the underlying principles of neural networks and could lead to more robust and reliable systems.

The work itself does not have any direct negative societal impacts. However, it is important to recognize that the broader research area of automated machine learning and neural architecture search could potentially have unintended negative consequences if not used responsibly.

Potential negative impacts: 1) our work could not directly resolve the risk that the AutoML algorithms may learn and replicate existing biases in the data, leading to biased decision-making in applications such as hiring, lending, and criminal justice; 2) our work could not directly resolve the risk that the automation of tasks that were previously performed by humans, leading to job displacement and potentially widening economic inequality.

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

## A  Experiment Settings

Tiny Imagenet contains 200 classes for training, each class has 500 images, and the test set contains 10,000 images. All images are 64×64 colored ones. Networks are trained for 3000 epochs with SGD, a batch size of 128, and a constant learning rate of 0.005. No augmentations, regularizations, weight decay, or momentum are applied. Layer width $m = 256$ for all networks, which is a typical choice studied in previous work (Lee et al., 2020). Note that on average, architectures in our architecture space are of 3.33M parameters (with a standard deviation of 0.076M), versus 0.1M images on Tiny-ImageNet. This means our networks are in an over-parameterized regime. We use the converged training loss as expressivity (lower the more expressive). Inspired by (Hanin and Rolnick, 2018), we measure the convergence as how many epochs a network requires to reach 50% accuracy (fewer epochs the faster convergence). The gap between test and training loss represents the generalization (smaller gaps generalize better).

Table 1: Ensembling + low-rank regularization can improve expressivity-convergence-generalization trade-off of architectures. "(·, ·)" indicates two architectures to ensemble. By ensembling two weak architectures ("II" and "III") with low-rank regularization (random unstructured pruning), we can achieve better trade-off ("V") than the best architecture ("I") in our space. Experiments done on Tiny-ImageNet (license is publicly available) and V100 GPUs.

| | Architecture | | | Ensemble | Pruning Ratios % | Rankings (out of 729 NNs, smaller the better) | | | Sum of Rankings |
|---|---|---|---|---|---|---|---|---|---|
| | $P$ | $\frac{1}{P}\sum_{p=1}^{P} d_p$ | Params. | | | Expressivity | Convergence | Generalization | |
| I. best of 729 NNs | 4 | 1.5 | 3.53M | | 0 | 71 | 76 | 5 | 152 |
| II. wide shallow | 4 | 1 | 3.46M | | 0 | 217 | **39** | **1** | 257 |
| III. deep narrow | 1 | 3 | 3.4M | | 0 | **1** | 538 | 541 | 1080 |
| IV. ensemble (II, III) | (4, 1) | (1, 3) | 3.66M | ✓ | 0 | 5 | 46 | 180 | 231 |
| V. ensemble (II, III) | (4, 1) | (1, 3) | 3.51M | ✓ | II: 20, III: 50 | 39 | 32 | 38 | **109** |

# B  Better Trade-off by Ensembling and Low-rank Regularization

One would feel discouraged about the "no free lunch" behavior of network architectures. Indeed, a single architecture cannot improve them all at the same time. However, this motivates us to further study: *given the winner of each aspect, can we integrate them into a stronger one with a better trade-off in all aspects?*

Our first intuition is to **ensemble multiple architectures** that can cover all three aspects. We choose a wide shallow architecture ($P = 4$, $\frac{1}{P}\sum_{p=1}^{P} d_p = 1$) for its convergence and the generalization (similar bias in architectures), and include another deep narrow one ($P = 1$, $\frac{1}{P}\sum_{p=1}^{P} d_p = 3$) for its expressivity. Inspired by the super-network concept in (Liu et al., 2018b), we make two architectures share their nodes (features) but with separated edges (keep their own weights). This will improve both convergence and expressivity, but will jeopardize its generalization (row "IV" in Table 1).

Regularizations, such as weight decay and augmentation, are introduced in Deep Learning to avoid overfitting. Since our focus is on network architecture, we instead seek implicit regularizations on the architecture itself, instead of from an optimization perspective. Implicit bias to low-rankness is observed in both deep linear networks (Arora et al., 2019a) and practical ones (Gur-Ari et al., 2018; Hu et al., 2021). Therefore, we explore **low-rank regularizations on our network architectures**.

Recent works demonstrate that pruning can act as an implicit regularization and remove the model's redundant intrinsic dimensions (Chen et al., 2020; Xu et al., 2019; Yu et al., 2017). Here we consider **unstructured random pruning**, to reduce a network's redundancy and regularize its expressivity. We multiply a binary mask to each weight $M \cdot W$, where $M$ is sparse at a certain pruning ratio. With an appropriate pruning ratio, in row "V" of Table 1 we show that, an ensemble of multiple (weak) architectures with low-rank regularizations can achieve a better trade-off among the three aspects, even higher than the best (single) architecture (row "I"). Specifically, we calculate three rankings of the expressivity, convergence, and generalization over 729 architectures in our architecture space ($H = 3$), and compare the sum of the three rankings. Our ensembling plus low-rank regularization methods achieve a better sum of rankings, with an even fewer number of parameters. For more ablation studies, please refer to our Appendix C.2 in the supplement.

# C  More experiments

## C.1  Empirical expressivity, convergence, generalization vs. graph topology on CIFAR-10/100

We include more experimental results here. First, we show correlations between expressivity, convergence, generalization and graph topologies on CIFAR-10 and CIFAR-100. Figure 5 again demonstrates that the expressivity favors deep wide networks, whereas the convergence and generalization prefer wide shallow networks.

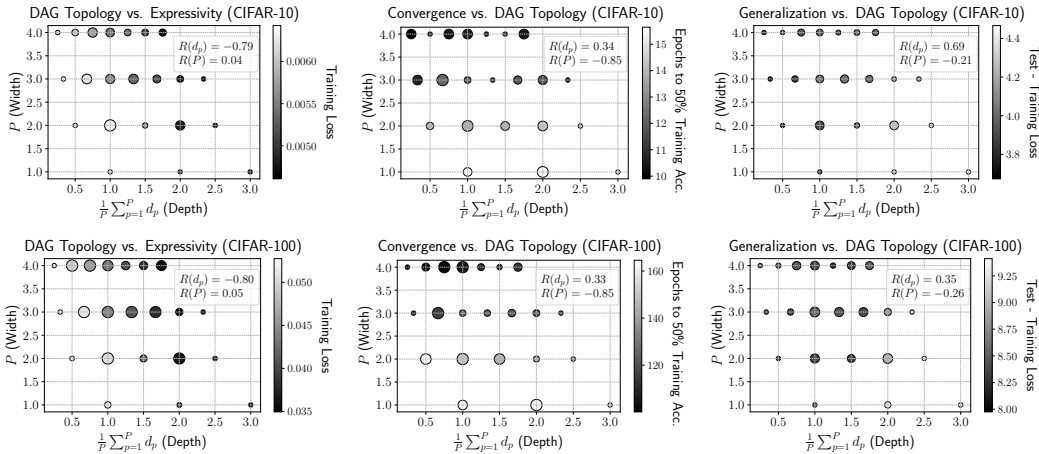

Figure 5: Deep narrow networks have higher expressivity (left), while wide shallow networks converge faster (middle) and generalize better (right). All 729 networks ($H = 3$) are trained on CIFAR-10 and CIFAR-100. Smaller values (dark circles) the better in all three plots. Kendall-tau correlations ("$R$") are reported in legends. Radius of circles indicates standard deviations over networks of the same graph topology ($P$ and $\frac{1}{P}\sum_{p=1}^{P} d_p$). **Left**: expressivity by training loss at convergence. **Middle**: convergence by number of epochs required to reach 50% training accuracy. **Right**: generalization gap between test and training loss at convergence.

We also verify the correlations between empirical metrics of expressivity/convergence/generalization versus theoretical ones. As shown in Figure 6, all three theoretical indicators are aligned with empirical metrics.

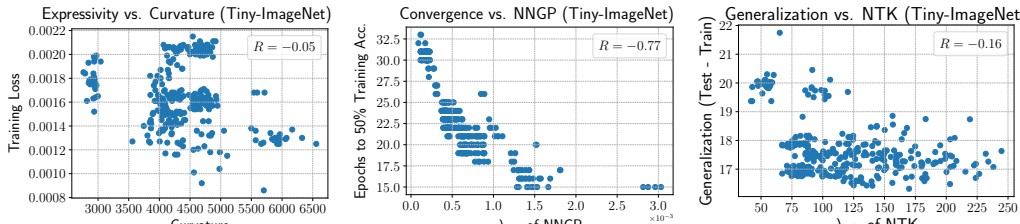

Figure 6: **Left**: for expressivity, high curvature indicates low training loss at convergence. **Middle**: for convergence, large $\lambda_{\min}$ of NNGP indicates fewer number of epochs required to reach 50% training accuracy. **Right**: for generalization, large $\lambda_{\min}$ of NTK indicates smaller gap between test and training loss at convergence. All 729 networks ($H = 3$) are trained on Tiny-ImageNet. Kendall-tau correlations ($R$) are reported in legends.

## C.2 Ablation study of ensembling and pruning

We conduct a systematic study of different pruning ratios on two ensembled graphs in Figure 7. architecture 1 (row II in Table 1) is wide shallow and contributes to the convergence and generalization (with poor expressivity). architecture 2 (row III in Table 1) is deep narrow and contributes to the expressivity (with poor convergence and generalization).

In Figure 7, lower (dark) sum of three rankings indicate better trade-off. We can see that more aggressive pruning ratios on architecture 2 (negative correlation with the sum of rankings) will mitigate its overfitting issue, improving its generalization and leading to better trade-off. In contrast, we should not impose heavy pruning ratios on architecture 1, which will further jeopardize its expressivity. Meanwhile, higher pruning ratios (top right) will lead to smaller model sizes (circle radiuses).

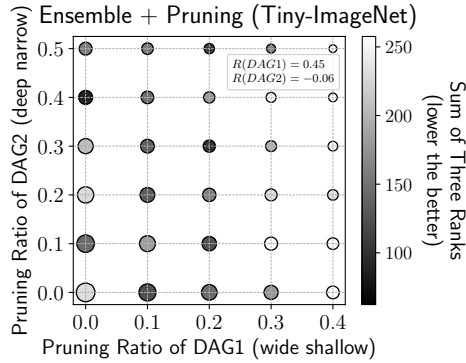

Figure 7: Different pruning ratios on two ensembled architectures. X-axis: wide shallow graph (row II in Tab. 1). Y-axis: deep narrow graph (row III in Tab. 1). Circle radiuses indicate model sizes (number of parameters) after pruning. Kendall-tau correlations between pruning ratios and the sum of rankings are reported in legend.

## C.3 Training with Optimal Learning Rates Tailored for Architectures

Standard architecture benchmarks (Ying et al., 2019; Dong and Yang, 2020) train networks with a shared training recipe. As architecture topologies are very diverse in these benchmarks, blindly using the same training setting for all networks may not be optimal. In this section, we further study the empirical expressivity, trainability, and generalization by training networks with their optimal learning rates tuned by grid search. As shown in Figure 8, even with architecture-wise optimal learning rates, our conclusion still holds: in a complete architecture space, given a fixed budget of network parameters, deep narrow networks have higher expressivity, while wide shallow networks converge faster and generalize better.

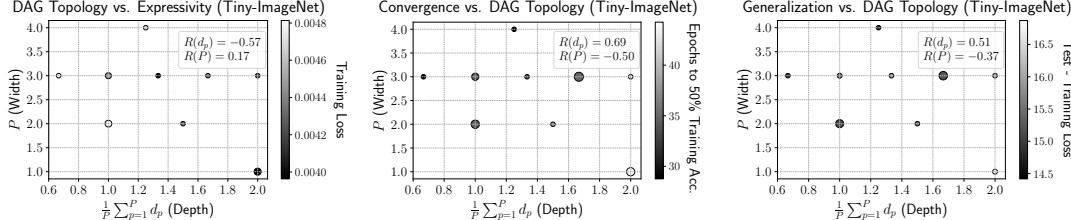

Figure 8: Deep narrow networks have higher expressivity (left), while wide shallow networks converge faster (middle) and generalize better (right). Networks ($H = 3$) are trained on Tiny-ImageNet, with optimal learning rates tuned for each network. Smaller values (dark circles) the better in all three plots. Kendall-tau correlations ("$R$") are reported in legends. Radius of circles indicates standard deviations over networks of the same graph topology ($P$ and $\frac{1}{P} \sum_{p=1}^{P} d_p$). **Left**: expressivity by training loss at convergence. **Middle**: convergence by number of epochs required to reach 50% training accuracy. **Right**: generalization gap between test and training loss at convergence.

## D  Expressivity

In this section, we study the functional complexity for deep networks. Our goal is to compare the expressivity of different networks and establish links to their graph topologies. Below, we consider a simple circle input $X^{(0)}(\theta) = \sqrt{N_0} \left[ \mathbf{u}_0 \cos(\theta) + \mathbf{u}_1 \sin(\theta) \right]$, where $\theta \in [0, 2\pi)$, $\mathbf{u}_0$ and $\mathbf{u}_1$ form an orthonormal basis for a 2 dimensional subspace of the input space $\mathbb{R}^{N_0}$ (e.g. $N_0 = 3 \times 32 \times 32$ for images in CIFAR-10 dataset).

### D.1  Proof of Theorem 4.1

Before we prove Theorem 4.1, we first give a general fact about the number of paths in a graph. Suppose a graph of nodes $0, 1, \cdots, H - 1$, it has $P_{H-1}$ end-to-end paths. Then, if we add one more

node $(H)$ to this graph, it will have $P_H = \sum_{h=0}^{H-1} \mathbb{1}(W^{(h,H)} \neq 0) \cdot P_h$ paths, where $\mathbb{1}(W^{(h,H)} \neq 0) = 1$ if $W^{(h,H)} \neq 0$ otherwise is 0. We set $P_0 = 1$. Intuitively, we can create a new edge from each previous node to the new node $H$. Therefore, for node $h \in [0, H-1]$, we first have a number of $P_h$ path choices to go from node 0 to $h$, then have one choice to go from $h$ to $H$. Therefore, $P_H$ simply equals to the sum of all previous paths, with any zero operators (disabled edges) removed.

**Theorem 4.1** (Jacobian in Architectures). *For ReLU networks in our architecture space (Figure 2) of nodes $0, 1, \cdots, H$. The total number of end-to-end paths is $P_H$, and the depth of each path is $d_p$ ($p = 1, \cdots, P_H$). Weights are initialized by the standard He normal initialization (He et al., 2015). The expectation (over the weight distribution) of the Jacobian's norm of this network is:*

$$\int_0^{2\pi} \mathbb{E}\left[\|J(\theta)\|\right] d\theta = C \cdot \sum_{p=1}^{P_H} \exp\left[-\frac{5}{8}\frac{d_p}{m} + O\left(\frac{d_p}{m^2}\right)\right], \tag{11}$$

*where $C = \dfrac{\Gamma\left(\frac{m+1}{2}\right)}{\Gamma\left(\frac{m}{2}\right)\left(\frac{m}{2}\right)^{1/2}}$, $m$ is the hidden layer width (Eq. 2), and $\Gamma(\cdot)$ denotes the Gamma function.*

*Proof of Theorem 4.1.* For a ReLU MLP of $L$ layers, based on the chain rule, we can write its Jacobian as

$$J_{X^{(0)}} = D^{(L)}W^{(L)}D^{(L-1)}W^{(L-1)} \cdots D^{(1)}W^{(1)}$$

where $W^{(\ell)}$ is the matrix of weights from layer $\ell - 1$ to layer $\ell$ and $D^{(\ell)}$ is an $m \times m$ diagonal matrix:

$$D^{(\ell)} = \mathrm{Diag}\left(\mathbb{1}_{\left\{z_i^{(\ell)} \geq 0\right\}}, i = 1, \ldots, m\right)$$

whose diagonal entries are 0 or 1 depending on whether the pre-activation $z_i^{(\ell)}$ of neuron $i$ in layer $\ell$ is positive at our fixed input.

Next, we use induction to show that the Jacobian of a network as $J_{X^{(0)}} = \sum_{p=1}^{P_H} \prod_{\ell=0}^{d_p^{(0,H)}} D^{(\ell)}W^{(\ell)}$, where $d_p^{(0,H)}$ indicates the depth of $p$-th path that starts from node 0 and ends at node $H$.

1. Suppose for a network of $H - 1$ nodes, it has $P_{H-1}$ paths and its Jacobian is $J_{X^{(0)}} = \sum_{p=1}^{P_{H-1}} \prod_{\ell=0}^{d_p^{(0,H-1)}} D^{(\ell)}W^{(\ell)}$.

2. Now we add one more node $H$. We can add one edge from each of previous nodes $0, \cdots, H-1$ to this new node $H$.

3. For example, for the node $H - 1$, the newly added edge contributes to the Jacobian with $\mathbb{1}(W^{(H-1,H)} \neq 0) \cdot (D^{(H-1,H)}W^{(H-1,H)})^{d^{(H-1,H)}}$, where the depth from node $H - 1$ to $H$ is $d^{(H-1,H)} = 1$ if this edge is a "linear + ReLU" layer, or $d^{(H-1,H)} = 0$ if this edge is a skip-connection.

4. Thus, after adding the new edge, the Jacobian from the node $H - 1$ is $J_{X^{(0)}} = \mathbb{1}(W^{(H-1,H)} \neq 0) \cdot \sum_{p=1}^{P_{H-1}} \left(\prod_{\ell=1}^{d_p^{(0,H-1)}} D^{\ell}W^{\ell}\right) (D^{(H-1,H)}W^{(H-1,H)})^{d^{(H-1,H)}}$. We can merge $d_p^{(0,H-1)}$ and $d^{(H-1,H)}$ to complete a new path that starts at node 0 and ends at node $H$, whose depth is $d_p^{(0,H)}$.

Now, consider all nodes $1, \cdots, H-1$, we can write the new Jacobian as:

$$\mathbf{J}_{X^{(0)}} = \tag{12}$$

$$\mathbb{1}(W^{(H-1,H)} \neq \mathbf{0}) \sum_{p=1}^{P_{H-1}} \left( \prod_{\ell=1}^{d_p^{(0,H-1)}} D^\ell W^\ell \right) (D^{(H-1,H)} W^{(H-1,H)})^{d^{(H-1,H)}} \quad \text{(node } H-1) \tag{13}$$

$$+ \mathbb{1}(W^{(H-2,H)} \neq \mathbf{0}) \sum_{p=1}^{P_{H-2}} \left( \prod_{\ell=1}^{d_p^{(0,H-2)}} D^\ell W^\ell \right) (D^{(H-2,H)} W^{(H-2,H)})^{d^{(H-2,H)}} \quad \text{(node } H-2) \tag{14}$$

$$+ \cdots \tag{15}$$

$$+ \mathbb{1}(W^{(1,H)} \neq \mathbf{0}) \sum_{p=1}^{P_1} \left( \prod_{\ell=1}^{d_p^{(0,1)}} D^\ell W^\ell \right) (D^{(1,H)} W^{(1,H)})^{d^{(1,H)}} \quad \text{(node 1)} \tag{16}$$

$$+ \mathbb{1}(W^{(0,H)} \neq \mathbf{0}) (D^{(0,H)} W^{(0,H)})^{d^{(0,H)}} \quad \text{(node 0)} \tag{17}$$

$$= \mathbb{1}(W^{(H-1,H)} \neq \mathbf{0}) \sum_{p=1}^{P_{H-1}} \left( \prod_{\ell=1}^{d_p^{(0,H-1)}+d_p^{(H-1,H)}} D^{(\ell)} W^{(\ell)} \right) \quad \text{(node } H-1) \tag{18}$$

$$+ \mathbb{1}(W^{(H-2,H)} \neq \mathbf{0}) \sum_{p=1}^{P_{H-2}} \left( \prod_{\ell=1}^{d_p^{(0,H-2)}+d_p^{(H-2,H)}} D^{(\ell)} W^{(\ell)} \right) \quad \text{(node } H-2) \tag{19}$$

$$+ \cdots \tag{20}$$

$$+ \mathbb{1}(W^{(1,H)} \neq \mathbf{0}) \sum_{p=1}^{P_1} \left( \prod_{\ell=1}^{d_p^{(0,1)}+d_p^{(1,H)}} D^{(\ell)} W^{(\ell)} \right) \quad \text{(node 1)} \tag{21}$$

$$+ \mathbb{1}(W^{(0,H)} \neq \mathbf{0}) (D^{(0,H)} W^{(0,H)})^{d^{(0,H)}} \quad \text{(node 0)} \tag{22}$$

$$= \sum_{p=1}^{\sum_{h=0}^{H-1} \mathbb{1}(W^{(h,H)} \neq \mathbf{0}) P_h} \left( \prod_{\ell=1}^{d_p^{(0,H)}} D^{(\ell)} W^{(\ell)} \right) \tag{23}$$

$$= \sum_{p=1}^{P_H} \prod_{\ell=1}^{d_p^{(0,H)}} D^{(\ell)} W^{(\ell)} \tag{24}$$

Next, based on the Proposition C.2 in (Hanin et al., 2021), we know that

$$\int_0^{2\pi} \mathbb{E} \| D^{(\ell)} W^{(\ell)} X^{(\ell-1)} \| d\theta = 1 - \frac{5}{8m} + O(m^{-2}),$$

where the expectation is taken over the weight distribution. Therefore, we can conclude:

$$\int_0^{2\pi} \mathbb{E} \left[ \| \mathbf{J}(\theta) \| \right] d\theta = \int_0^{2\pi} \sum_{p=1}^{P_H} \prod_{\ell=1}^{d_p^{(0,H)}} \mathbb{E} \| D^{(\ell)} W^{(\ell)} X^{(\ell-1)} \| d\theta$$

$$= C \cdot \sum_{p=1}^{P_H} \exp\left[ -\frac{5}{8} \frac{d_p}{m} + O\left( \frac{d_p}{m^2} \right) \right], \tag{25}$$

where $C = \frac{\Gamma\left(\frac{m+1}{2}\right)}{\Gamma\left(\frac{m}{2}\right)\left(\frac{m}{2}\right)^{1/2}}$, $m$ is the hidden layer width (Eq. 2), and $\Gamma(\cdot)$ denotes the Gamma function.

$\square$

## D.2 Proof of Corollary 4.2

We first give the general definition of manifold curvature.

**Lemma D.1** (Curvature of Curves (Lee, 2006)). *Consider a curve in Riemannian manifold $M$, that is a map $X^{(H)}(\theta) : I \rightarrow M$, where $I \subset \mathbb{R}$ is some interval. Define $\mathbf{J}(\theta) = \partial_\theta X^{(H)}(\theta)$, i.e. the input-output Jacobian of the map $X^{(H)}$. We have the curvature of the curve as*

$$\kappa(\theta) = \|\mathbf{J}(\theta)\|^{-3}\sqrt{\|\mathbf{J}(\theta)\|^2\|\partial_\theta \mathbf{J}(\theta)\|^2 - (\mathbf{J}(\theta) \cdot \partial_\theta \mathbf{J}(\theta))^2}. \tag{26}$$

Lemma D.1 is also used in (Poole et al., 2016) to characterize the expressivity of networks.

**Corollary 4.2** (Curvature and Jacobian). *For a ReLU network, we have its curvature as the reciprocal of the norm of the input-output Jacobian:*

$$\kappa(\theta) = \|\mathbf{J}(\theta)\|^{-1} \tag{27}$$

*Proof of Corollary 4.2.* Note that we have $\mathbf{J}_{X^{(0)}} = \prod_{\ell=1}^{L} D^{(\ell)} W^{(\ell)}$ and $\partial_\theta \mathbf{J}_{X^{(0)}} = 0$. Therefore:

$$\begin{aligned}
\mathbf{J}_\theta &= \mathbf{J}_{X^{(0)}} \cdot \frac{\partial X^{(0)}}{\partial \theta} = \sqrt{N_0} \left[ -\mathbf{u}_0 \sin(\theta) + \mathbf{u}_1 \cos(\theta) \right] \cdot \mathbf{J}_{X^{(0)}} \\
\partial_\theta \mathbf{J}_\theta &= \frac{\partial \mathbf{J}_{X^{(0)}}}{\partial \theta} \cdot \frac{\partial X^{(0)}}{\partial \theta} + \mathbf{J}_{X^{(0)}} \cdot \frac{\partial^2 X^{(0)}}{\partial \theta^2} \\
&= 0 \cdot \frac{\partial X^{(0)}}{\partial \theta} + \sqrt{N_0} \left[ -\mathbf{u}_0 \cos(\theta) - \mathbf{u}_1 \sin(\theta) \right] \cdot \mathbf{J}_{X^{(0)}} \\
&= -\sqrt{N_0} \left[ \mathbf{u}_0 \cos(\theta) + \mathbf{u}_1 \sin(\theta) \right] \cdot \mathbf{J}_{X^{(0)}}
\end{aligned} \tag{28}$$

Given $\mathbf{u}_0$ and $\mathbf{u}_1$ are orthonormal bases, and based on the definition in D.1, we have:

$$\begin{aligned}
\kappa(\theta) &= \|\mathbf{J}(\theta)\|^{-3}\sqrt{\|\mathbf{J}(\theta)\|^2\|\partial_\theta \mathbf{J}(\theta)\|^2 - (\mathbf{J}(\theta) \cdot \partial_\theta \mathbf{J}(\theta))^2} \\
&= \|\mathbf{J}(\theta)\|^{-3}\sqrt{N_0\|\mathbf{J}_{X^{(0)}}\|^2 \cdot N_0\|\mathbf{J}_{X^{(0)}}\|^2 - (N_0(\sin(\theta)\cos(\theta) - \cos(\theta)\sin(\theta))\|\mathbf{J}_{X^{(0)}}\|^2)^2} \\
&= \|\mathbf{J}(\theta)\|^{-1}
\end{aligned} \tag{29}$$

$\square$

## E Convergence

We first follow the definition of $f$ in (Chen et al., 2022) that characterizes how NNGP kernel propagates through ReLU layers:

**Lemma E.1** (Propagation of $K$ (Chen et al., 2022)). *Let ReLU activation $\sigma(x) = \max\{0, x\}$ and $c_\sigma = 2$. Define the propagation as $K^{(l)} = f(K^{(l-1)})$ and $b^{(l)} = g(b^{(l-1)})$. When the edge operation is a linear transformation, we have:*

$$\begin{aligned}
K_{ii}^{(l)} &= f(K_{ii}^{(l-1)}) = K_{ii}^{(l-1)} \\
K_{ij}^{(l)} &= f(K_{ij}^{(l-1)}) = h(C_{ij}^{(l-1)})\sqrt{K_{ii}^{(l)} K_{jj}^{(l)}} \\
&= \frac{\dfrac{2C_{ij}^{(l-1)} \arcsin C_{ij}^{(l-1)} + 2\sqrt{1 - (C_{ij}^{(l-1)})^2} + \pi C_{ij}^{(l-1)}}{2\pi}}{\sqrt{K_{ii}^{(l)} K_{jj}^{(l)}}} \\
C_{ij}^{(l)} &= K_{ij}^{(l)} / \sqrt{K_{ii}^{(l)} K_{jj}^{(l)}} \\
b_i^{(l)} &= g(b_i^{(l-1)}) = \frac{\sqrt{c_\sigma}}{2}.
\end{aligned} \tag{30}$$

We re-state some facts:

- $h(\cdot)$ is a monotonically increasing function in $[0, 1)$, and $\lim_{C_{ij}^{(l-1)} \to 1-} h(C_{ij}^{(l-1)}) = 1$ (Hayou et al., 2019).

- $f^{d_p}(K_{ij}^{(0)})$ reflects how NNGP propagates through a path, and $f(K_{ij}^{(l-1)}) > K_{ij}^{(l-1)}$.

### E.1 Proof of Theorem 4.4

We first give a condition where the propagation of the sum of multiple NNGP kernels equals to the sum of individual propagations of NNGP kernels.

**Corollary E.1.** *Given a graph of nodes $0, 1, \cdots, H$, assume $\forall p_1, p_2 \in [1, P_H], p_1 \neq p_2$, we have $d_{p_1} = d_{p_2}$, then we have:*

$$f(\sum_{p=1}^{P_H} f^{d_p}(K^{(0)})) = \sum_{p=1}^{P_H} f(f^{d_p}(K^{(0)})). \tag{31}$$

*Proof of Corollary E.1.* Denote $d = d_p$ for $p \in [1, P_H]$. Based on Lemma F.4, we have:

$$
\begin{aligned}
f(\sum_{p=1}^{P_H} f^{d_p}(K_{ij}^{(0)})) &= f(P_H \cdot f^d(K_{ij}^{(0)})) \\
&= h(\frac{P_H \cdot f^d(K_{ij}^{(0)})}{\sqrt{P_H \cdot f^d(K_{ii}^{(0)}) \cdot P_H \cdot f^d(K_{jj}^{(0)})}}) \sqrt{f(P_H \cdot f^d(K_{ii}^{(0)})) \cdot f(P_H \cdot f^d(K_{jj}^{(0)}))} \\
&= h(\frac{f^d(K_{ij}^{(0)})}{\sqrt{f^d(K_{ii}^{(0)}) \cdot f^d(K_{jj}^{(0)})}}) \sqrt{P_H \cdot P_H} \\
&= P_H \cdot f(f^d(K_{ij}^{(0)})) = \sum_{p=1}^{P_H} f(f^{d_p}(K_{ij}^{(0)}))
\end{aligned}
\tag{32}
$$

$\square$

*Remark E.2.* Below, we assume that all paths that end at any intermediate node have the same depth (i.e., have the same number of ReLU layer), where the Corollary E.1 will hold at any intermediate node $h \in [0, H]$.

**Theorem 4.4** ($\lambda_{\min}(K^{(H)})$ of Architectures). *For ReLU networks in our architecture space (Figure 2) of nodes $0, 1, \cdots, H$. The total number of end-to-end paths is $P_H$, and the depth of each path is $d_p$ ($p = 1, \cdots, P_H$). The least eigenvalue of NNGP kernel of this network is:*

$$\lambda_{\min}(K^{(H)}) \leq \min_{i \neq j} \left[ P_H - \sum_{p=1}^{P_H} f^{d_p}(K_{ij}^{(0)}) \right] \qquad i, j \in [1, N]. \tag{33}$$

*Proof of Theorem 4.4.* We adopt the similar idea in our proof in Section D.1.

We first use induction to prove that given a graph of nodes $0, 1, \cdots, H$, its NNGP kernel at $X^{(H)}$ is:

$$K^{(H)} = \sum_{p=1}^{P_H} f^{d_p^{(0,H)}}(K^{(0)}).$$

1. Suppose for a network of $H - 1$ nodes, its NNGP kernel at $X^{(H-1)}$ has $K^{(H-1)} = \sum_{p=1}^{P_{H-1}} f^{d_p^{(0,H-1)}}(K^{(0)})$.

2. Now we add one more node $H$. We can add one edge from each of previous nodes $0, \cdots, H-1$ to this new node $H$.

3. For example, for the node $H-1$, the newly added edge contributes to the NNGP kernel with $\mathbb{1}(W^{(H-1,H)} \neq 0) \cdot f^{d^{(H-1,H)}}(K^{(H-1)})$, where the depth from node $H-1$ to $H$ is $d^{(H-1,H)} = 1$ if this edge is a "linear + ReLU" layer, or $d^{(H-1,H)} = 0$ if this edge is a skip-connection (we define $f^0(K) = 1$).

4. Thus, after adding the new edge, the NNGP kernel from the node $H-1$ to node $H$ is $K^{(H)} = \mathbb{1}(W^{(H-1,H)} \neq 0) \cdot f^{d^{(H-1,H)}}\left(\sum_{p=1}^{P_{H-1}} f^{d_p^{(0,H-1)}}(K^{(0)})\right)$.

5. Based on Corollary E.1, we have:

$$K^{(H)} = \mathbb{1}(W^{(H-1,H)} \neq 0) \cdot \sum_{p=1}^{P_{H-1}} f^{d^{(H-1,H)}}\left(f^{d_p^{(0,H-1)}}(K^{(0)})\right).$$

6. We can merge $d_p^{(0,H-1)}$ and $d^{(H-1,H)}$ to complete a new path that starts at node 0 and ends at node $H$, whose depth is $d_p^{(0,H)}$. Therefore, we have $K^{(H)} = \mathbb{1}(W^{(H-1,H)} \neq 0) \cdot \sum_{p=1}^{P_{H-1}} f^{d_p^{(0,H)}}(K^{(0)})$.

Now, consider all nodes $1, \cdots, H-1$, we can write the new NNGP kernel as:

$$K^{(H)} = \mathbb{1}(W^{(H-1,H)} \neq 0) \sum_{p=1}^{P_{H-1}} f^{d_p^{(0,H)}}(K^{(0)})(\text{node } H-1) \tag{34}$$

$$+ \mathbb{1}(W^{(H-2,H)} \neq 0) \sum_{p=1}^{P_{H-2}} f^{d_p^{(0,H)}}(K^{(0)}) \, (\text{node } H-2) \tag{35}$$

$$+ \cdots \tag{36}$$

$$+ \mathbb{1}(W^{(1,H)} \neq 0) \sum_{p=1}^{P_1} f^{d_p^{(0,H)}}(K^{(0)}) \qquad (\text{node } 1) \tag{37}$$

$$+ \mathbb{1}(W^{(0,H)} \neq 0) f^{d^{(0,H)}}(K^{(0)}) \qquad (\text{node } 0) \tag{38}$$

$$= \sum_{p=1}^{\sum_{h=0}^{H-1} \mathbb{1}(W^{(h,H)} \neq 0)P_h} \left(f^{d_p^{(0,H)}}(K^{(0)})\right) \tag{39}$$

$$= \sum_{p=1}^{P_H} f^{d_p^{(0,H)}}(K^{(0)}) \tag{40}$$

Note that for diagonal elements (inner product of features from the same sample), since $K_{ii}^{(0)} = 1$ and $f^d(K_{ii}^{(0)}) = 1$ for any $d > 0$, we have $K_{ii}^{(H)} = P_H$ ($i \in [1, N]$). Therefore, we have:

$$\lambda_{\min}(K_{2\times2}^{(H)}) = \min_{i \neq j} P_H - K_{ij}^{(H)}$$

$$= \min_{i \neq j} \left[ P_H - \sum_{p=1}^{P_H} f^{d_p^{(0,H)}}(K_{ij}^{(0)}) \right] \qquad i, j \in [1, N], \tag{41}$$

where $K_{2\times2}^{(H)}$ denotes any $2 \times 2$ submatrix of $K^{(H)}$. From Lemma 3.3 in (Chen et al., 2022), we also have $\lambda_{\min}(K) \leq \min_{i \neq j} \lambda_{\min}(K_{2\times2})$. This complete the proof. □

## F  Generalization

### F.1  Proof of Theorem 4.5

*Proof.* Here we provide a high-level proof for the generalization bound of architecture. Before giving the proof of Theorem 4.5, we first introduce several lemmas inspired by Cao and Gu (2019).

**Lemma F.1.** *There exists an absolute constant $\kappa$ such that, with probability at least $1 - O(nL^2) \cdot \exp[-\Omega(m\omega^{2/3}L)]$ over the randomness of $W^{(1)}$, where $L = \sum_{p=1}^{P_H} d_p$, for all $i \in [n]$ and $W, W' \in \mathcal{B}(W^{(1)}, \omega)$ with $\omega \leq \kappa L^{-6}[\log(m)]^{-3/2}$, it holds uniformly that*

$$|f_{W'}(x_i) - f_W(x_i) - \langle \nabla f_W(x_i), W' - W \rangle| \leq O\left(\omega^{1/3}L^2\sqrt{m\log(m)}\right) \cdot \sum_{l=1}^{L-1} \|W^{(l')} - W^{(l)}\|_2.$$

Note that $L = \sum_{p=1}^{P_H} d_p$ considers all the operations of trainable parameters. Since the cross-entropy loss $\ell(\cdot)$ is convex, given Lemma F.1, we can show in the following lemma that near initialization, $L_i(W)$ is also almost a convex function of $W$ for any $i \in [n]$.

**Lemma F.2.** *There exists an absolute constant $\kappa$ such that, with probability at least $1 - O(nL^2) \cdot \exp[-\Omega(m\omega^{2/3}L)]$ over the randomness of $W^{(1)}$, for any $\epsilon > 0$, $i \in [n]$ and $W, W' \in \mathcal{B}(W^{(1)}, \omega)$ with $\omega \leq \kappa L^{-6}m^{-3/8}[\log(m)]^{-3/2}\epsilon^{3/4}$, it holds uniformly that*

$$L_i(W') \geq L_i(W) + \langle \nabla_W L_i(W), W' - W \rangle - \epsilon.$$

We then derive a bound of the cumulative loss. The result is given in the following lemma.

**Lemma F.3.** *For any $\epsilon, \delta, R > 0$, there exists*

$$m^*(\epsilon, \delta, R, L) = \tilde{O}\big(\text{poly}(R, L)\big) \cdot \epsilon^{-14} \cdot \log(1/\delta)$$

*such that if $m \geq m^*(\epsilon, \delta, R, L)$, then with probability at least $1 - \delta$ over the randomness of $W^{(1)}$, for any $W^* \in \mathcal{B}(W^{(1)}, Rm^{-1/2})$, $W^{(1)}, \ldots, W^{(T)}$ with $\eta = \nu\epsilon/(Lm)$, $n = L^2R^2/(2\nu\epsilon^2)$ for some small enough absolute constant $\nu$ has the following cumulative loss bound:*

$$\sum_{i=1}^{n}L_i(W^{(i)}) \leq \sum_{i=1}^{n}L_i(W^*) + 3n\epsilon.$$

With the above lemmas at hand, we can apply Corollary 3.10. in (Cao and Gu, 2019), which states that: For any $\delta \in (0, e^{-1}]$, there exists $\tilde{m}(\delta, L, n, \lambda_{\min}(G^{(H)})) = \tilde{O}(\text{poly}(L, 1/\lambda_{\min}(G^{(H)})))n^7\log(1/\delta)$ that only depends on $\delta, L, n$ and $\lambda_{\min}(G^{(H)})$, such that if $m \geq \tilde{m}(\delta, L, n, \lambda_{\min}(G^{(H)}))$. Then with probability at least $1 - \delta$ over the randomness of $W$, the output of SGD with step size $\eta = \kappa \cdot \sqrt{y^\top(G^{(H)})^{-1}y/(m\sqrt{n})}$ for some small enough absolute constant $\kappa$ satisfies,

$$L_{\mathcal{D}}(f) \leq \tilde{O}\left(L \cdot \sqrt{\frac{y^\top(G^{(H)})^{-1}(X, X)y}{n}}\right) + O\left(\sqrt{\frac{\log(1/\delta)}{n}}\right) \tag{42}$$

where $L = \sum_{p=1}^{P_H} d_p$ is the total depth in a graph, $G^{(H)}$ denotes the NTK of the architecture. We use $\tilde{O}(\cdot)$ to hide the logarithmic factors in $O(\cdot)$.

We can apply the above bound to the graph structure, in which the depth is defined as the number of linear transform operations, i.e., $L = \sum_{p=1}^{P_H} d_p$, and $R = \sqrt{y^\top(G^{(H)})^{-1}(X, X)y}$. Therefore, we can achieve the following final result:

$$L_{\mathcal{D}}(f) \leq \tilde{O}\left(\left[\left(\sum_{p=1}^{P_H} d_p\right) \cdot \sqrt{\frac{y^\top(G^{(H)})^{-1}(X, X)y}{n}}\right]\right) + O\left(\sqrt{\frac{\log(1/\delta)}{n}}\right). \tag{43}$$

$\square$

## F.2 Propagation of NTK of Architectures

In the last section, we present a generalization bound of an architecture through NTK, here we illustrate how to obtain the NTK recursively and provide the essential definitions.

The definition of NTK of an architecture is given as follows:

$$G^{(H)} = \left\langle \frac{\partial X^{(H)}}{\partial W}, \frac{\partial X^{(H)}}{\partial W} \right\rangle \tag{44}$$

where $W$ is the collection of all weights. Then we can recursively compute the NTK from the NNGP (Jacot et al., 2018; Arora et al., 2019c):

$$G^{(h)} = K^{(h)} + \dot{K}^{(h)} G^{(h-1)} \quad (h \in [1, H]), \qquad G^{(0)} = K^{(0)} \tag{45}$$

where $\dot{K}^{(h)} = \langle \dot{\rho}(W^{(h-1,h)} X^{(h-1)}), \dot{\rho}(W^{(h-1,h)} X^{(h-1)}) \rangle$.

By Eq. 45, we can recursively compute the NTK $G^{(H)}$ from $K^{(0)}$ through $K$ and $\dot{K}$.

Finally, we provide the definition of $\dot{f}$ given in the main text by the follow lemma.

**Lemma F.4** (Propagation of $\dot{K}$). *Let ReLU activation $\sigma(x) = \max\{0, x\}$ and $c_\sigma = 2$. Define the propagation as $\dot{K}^{(l)} = \dot{f}(K^{(l-1)})$. When the edge operation is a linear transformation, we have:*

$$\dot{K}_{ii}^{(l)} \equiv \dot{f}(K_{ii}^{(l-1)}) = 1$$

$$\dot{K}_{ij}^{(l)} = \dot{f}(K_{ij}^{(l-1)}) = \frac{\arcsin C_{ij}^{(l-1)}}{\pi} + \frac{1}{2} \tag{46}$$

$$C_{ij}^{(l)} = K_{ij}^{(l)} / \sqrt{K_{ii}^{(l)} K_{jj}^{(l)}}$$

*Proof.* According to NNGP propagation formulation (Lee et al., 2017), we have

$$K_{ii}^{(l)} = \int c_\sigma \mathcal{D}_z \sigma(\sqrt{K_{ii}^{(l)}} z)$$

$$K_{ij}^{(l)} = \int c_\sigma \mathcal{D}_{z_1} \mathcal{D}_{z_2} \sigma(u) \sigma(v) \tag{47}$$

where $u = \sqrt{K_{ii}^{(l-1)}} z_1$ and $v = \sqrt{K_{jj}^{(l-1)}} \left( C_{ij}^{(l-1)} z_1 + \sqrt{1 - (C_{ij}^{(l-1)})^2} z_2 \right)$, with $C_{ij}^{(l)} = K_{ij}^{(l)} / \sqrt{K_{ii}^{(l)} K_{jj}^{(l)}}$, where $z_1$ and $z_2$ are independent differential variables. Besides, $\int \mathcal{D}_z = \frac{1}{\sqrt{2\pi}} \int dz e^{-\frac{1}{2} z^2}$ is the measure for a normal distribution.

Then we take the condition that $\sigma(x) = \max\{0, x\}$ and $c_\sigma = 2$ into equations above and obtain

$$\dot{K}_{ii}^{(l)} = \int c_\sigma \mathcal{D}_z \dot{\sigma}^2(\sqrt{K_{ii}^{(l-1)}} z) = \int_0^\infty 2 \mathcal{D}_z = 1 \tag{48}$$

Besides, for diagonal elements, we have

$$C_{ij}^{(l)} \equiv h(C_{ij}^{(l-1)}) = \frac{2 C_{ij}^{(l-1)} \arcsin C_{ij}^{(l-1)} + 2\sqrt{1 - (C_{ij}^{(l-1)})^2} + \pi C_{ij}^{(l-1)}}{2\pi} \tag{49}$$

It is known that $f$ is differentiable and satisfies,

$$\dot{f}(K_{ij}^{(l-1)}) = \dot{h}(C_{ij}^{(l-1)}) = \frac{1}{\pi} \arcsin(C_{ij}^{(l-1)}) + \frac{1}{2} \tag{50}$$

$\square$

The lemma above provides the definition for $\dot{f}$, and indicates that $\dot{f}(K_{ij}) \in [0, 1)$ given $K_{ij} \in [0, 1)$.

### F.3 Proof of Theorem 4.6

**Theorem 4.6** ($\lambda_{\min}(G^{(H)})$ of Architectures). *For ReLU networks in our architecture space (Figure 2) of nodes $0, 1, \cdots, H$. The total number of end-to-end paths is $P_H$, and the depth of each path is $d_p$ ($p = 1, \cdots, P_H$). The least eigenvalue of NTK kernel of this network is:*

$$\lambda_{\min}(G^{(H)}) \leq \min_{i \neq j} \left[ P_H - \sum_{p=1}^{P_H} f^{d_p}(K_{ij}^{(0)}) + \sum_{p=1}^{P_H} d_p - \sum_{p=1}^{P_H} \sum_{e=1}^{d_p} f^e(K_{ij}^{(0)}) \prod_{k=1}^{e} \dot{f}^k(K_{ij}^{(0)}) \right] \quad i, j \in [1, N]. \tag{51}$$

*Proof.* In the proof, we follow Corollary E.1. According to the propagation function presented in Section F.2, we know that NTK at $X^{(H)}$ is based on NNGP:

$$G^{(H)} = K^{(H)} + \dot{K}^{(H)} G^{(H-1)} = \sum_{p=1}^{P_H} f^{d_p^{(0,H)}}(K^{(0)}) + \dot{K}^{(H)} G^{(H-1)} \tag{52}$$

The first equation is expanded from the recursive expression for NTK and the second equation is obtained by plugging the result for $K^{(H)}$ in the proof of Theorem 4.4. Note that the first $\sum_{p=1}^{P_H} f^{d_p^{(0,H)}}(K^{(0)})$ is fixed when the parameters of the architecture $\{d_p, P_H\}$ are given. However, the second term varies with the specific configuration of the architecture. In particular, the explicit form of $\lambda_{\min}(\dot{K}^{(H)} G^{(H-1)})$ will vary in different combinations of skip connections and "linear + ReLU" layers.

The next step is to calculate the exact contribution of the second term in the recursive formula Eq. 52. Our goal is to find an upper bound for the smallest eigenvalue of all graph structures that satisfy the given parameters $\{d_p, P_H\}$. Intuitively, to preserve more contributions from $\dot{K}^{(h)} G^{(h-1)}$ ($h \in [1, H]$), we need to put "linear + ReLU" operation in deeper layers, while leaving skip connections in shallower layers. To this end, we follow a proof strategy from deep to early layers in an architecture:

1. According to Lemma F.4, we have $\dot{K}_{ii}^{(H)} = 1 > \dot{K}_{ij}^{(H)} > 0$, which implies that $\lambda_{\min}((\dot{K}^{(H)} G^{(H-1)})_{2\times2}) > 0$ when the edge in an architecture is a "linear + ReLU" layer, otherwise $\dot{K}^{(H)} = 0$. Thus, to maximize the $\lambda_{\min}((\dot{K}^{(H)} G^{(H-1)})_{2\times2})$, the edge from node $H-1$ to node $H$ should be a "linear + ReLU" layer.

2. Now consider $G^{H-1} = K^{H-1} + \dot{K}^{H-1} G^{H-2} = \sum_{p=1}^{P_{H-1}} f^{d_p^{(0,H-1)}}(K^{(0)}) + \dot{K}^{H-1} G^{H-2}$. Again, to maximize the $\lambda_{\min}((\dot{K}^{(H-1)} G^{(H-2)})_{2\times2})$, the edge from node $H-1$ to node $H$ should be "linear + ReLU" layer.

3. Repeat the above step $d_p - 1$ times until $G^{H-d_p+1}$, which satisfies $G^{H-d_p+1} = K^{H-d_p+1} + \dot{K}^{H-d_p+1} G^{H-d_p} = \sum_{p=1}^{P_{H-d_p+1}} f^{d_p^{(0,H-1)}}(K^{(0)}) + \dot{K}^{H-d_p+1} G^{H-d_p}$. Again, to maximize the $\lambda_{\min}((\dot{K}^{(H-d_p+1)} G^{(H-d_p)})_{2\times2})$, the edge from node $H-d_p$ to node $H-d_p+1$ should be "linear + ReLU" layer.

As a result, we obtain the structure of the architecture that can maximize the smallest eigenvalue of NTK. Finally, we have

$$\lambda_{\min}((\dot{K}^{(H)} G^{(H-1)})_{2\times2}) \leq \sum_{p=1}^{P_H} d_p - \sum_{p=1}^{P_H} \sum_{e=1}^{d_p} f^e(K_{ij}^{(0)}) \prod_{k=1}^{e} \dot{f}^k(K_{ij}^{(0)}) \tag{53}$$

The first term of right hand side is from diagonal element of $\dot{K}^{(H)} G^{(H-1)}$ and second term of right hand side comes from non-diagonal element of $\dot{K}^{(H)} G^{(H-1)}$, where there is $d_p$ terms for summation

and the product is due to the recursive formula. Plug this result back to Eq. 52, we have

$$\lambda_{\min}(G_{2\times2}^{(H)}) \le P_H - \sum_{p=1}^{P_H} f^{d_p}(K_{ij}^{(0)}) + \sum_{p=1}^{P_H} d_p - \sum_{p=1}^{P_H} \sum_{e=1}^{d_p} f^e(K_{ij}^{(0)}) \prod_{k=1}^{e} \dot{f}^k(K_{ij}^{(0)}) \tag{54}$$

From Lemma 3.3 in (Chen et al., 2022), we finally have,

$$\lambda_{\min}(G^{(H)}) \le \min_{i \ne j} \left[ P_H - \sum_{p=1}^{P_H} f^{d_p}(K_{ij}^{(0)}) + \sum_{p=1}^{P_H} d_p - \sum_{p=1}^{P_H} \sum_{e=1}^{d_p} f^e(K_{ij}^{(0)}) \prod_{k=1}^{e} \dot{f}^k(K_{ij}^{(0)}) \right] \quad i, j \in [1, N]. \tag{55}$$

$\square$

