# OpenReview forum: "“No Free Lunch” in Neural Architectures? A Joint Analysis of Expressivity, Convergence, and Generalization"
_automl.cc/AutoML/2023/Conference — AutoML 2023 MainTrack_

### Official Review · Reviewer_VWUy · 2023-04-11

**Potential Impact On The Field Of Automl Rating:** 3
**Technical Quality And Correctness Rating:** 2
**Clarity Rating:** 3
**Actions Required To Increase Overall Recommendation:** (already addressed in depth above)

**Summary Of Contributions:**

This paper discusses the factors that define the "goodness" of neural architectures. It demonstrates that there there is no single architecture that best optimizes all 3 key properties, namely $\textbf{expressivity}$ (the complexity of functions a network can approximate), $\textbf{convergence}$ (training speed under gradient descent), and $\textbf{generalization}$ (performance on unseen data).

**Clarity:**

The structure of the paper is clear. However, apart from the technical discrepancies, the following points should further be clarified:
1)  It should be clear that the experiments are conducted in the domain of image recognition only. It should also be clear why a linear transformation with ReLU activation has been chosen together with skip connections to form the operation set, but other operations such as convolutions have been omitted.
2) Line 116: “skip-connection can improve the number of neurons from exponential to polynomial of the network depth.” What does improving the number of neurons mean?
3) Lemma 4.1 merely is used to proof corollary 4.2 in the appendix. Thus, it would be sufficient to state Lemma 4.2 in the appendix only. However, it is not trivial that “Theorem 4.1 indicates that networks of more short paths will more likely have a larger norm of input-output Jacobian”. This, on the other hand, could be elaborated on in more detail in the main paper.
4) There are some grammatical errors present throughout the paper that should be addressed and corrected.







**Overall Review:**

The paper gives a clear introduction to the topic and provides a clear structure of all findings. The experiments are limited in the sense that the width and depth of a neural architecture might be misinterpreted and the the relatively small operation set might influence the expressiveness of the architecture space. Also, some architectural topologies have been evaluated more intensively than others and there might be an unequal advantage from the pre-defined hyperparameter setting. The formal results offer valuable insights; however, it would be beneficial to present them as concisely as possible, emphasizing the essential conclusions only.

**Potential Impact On The Field Of Automl:**

This work provides empirical observations on the influence of the width and depth of a neural network on its expressivity, convergence, and generalisation. Also, it provides theoretical insights on the relation of the network topology with (1) the manifold curvature to determine the expressivity, (2) the least eigenvalue of the Neural Network Gaussian Process (NNGP) kernel to determine the convergence rate, and (3) the least eigenvalue of the Neural Tangent Kernel (NTK) and the generalisation gap.

However, the relationships between the test accuracy and the three properties does not exhibit a clear and direct connection. A comprehensive discussion on the choice of these specific properties is warranted. Also, other factors such as the predefined operation set, the hyperparameter setting, or the dataset properties (experiments are limited to image recognition) have not been analyzed, although they may play an important role for the performance comparison of neural networks and for drawing general conclusions. Moreover, the expressiveness of the empirical results is limited, as addressed below. Thus, currently there only is a small to medium potential impact.

**Review Confidence:**

4: You are confident in your assessment, but not absolutely certain. It is unlikely, but not impossible, that you did not understand some parts of the submission or that you are unfamiliar with some pieces of related work.

**Review Rating:**

4: Weak Reject: For instance, a paper with minor technical flaws, limited impact, and/or weak evaluation.

**Review Summary:**

Considering the technical inconsistencies and the limited impact, I am inclined to reject the paper in its present form. Nonetheless, if the identified shortcomings are adequately addressed and resolved, the work has the potential to provide valuable insights for the community.

**Technical Quality And Correctness:**

$\textbf{Definitions}$:

W is defined as “layer”, which might be misleading. For instance, in the case of the ResNet block depicted in Figure 2, $X^{(3)} = RELU(X^{(2)} W^{(2,3)}) + X^{(0)}$, which by this definition would be two layers. This has direct implications on the defined notion of "topological depth", as the average depth of this residual block is $\sum_{p} d_p = 1.5$, while the same topology without the residual connection has an average depth of 3 (see MLP in Figure 2).

$\textbf{Experimental setup:}$

In total, the architecture space of H = 3 only includes seven architectures with width P = 1 (only a single (!) architecture with depth d = 3, three architectures with d = 2, and three architectures with d = 1). Thus, the comparability of the reported standard deviations over the networks with the same graph topology is limited. Instead of reporting the results of all 729 architectures in the architecture space, one could pick the different graph topologies in this space and train the underlying architectures with different random seeds to get comparable results. Also, for each different topology, the best hyperparameters should be searched for, as a “comparison under the same training protocol may be unfair”, which is also stated in the limitations.

$\textbf{Interpretation of the results:}$
1) $\textit{“networks of deep and narrow topology have higher expressivity”}$: the training loss over all widths seems to be small if the depth is sufficiently big. So, the only observation that can be made is that the deeper the network, the higher the expressivity. Also, the training loss is relatively low for every converged network no matter the topology (difference of < 1e-4), implying that any conclusion about their expressivity might be limited.
2) $\textit{“wide and shallow ones show faster convergence”}$: exiting research has already shown that wide and shallow networks can have simpler optimization landscapes, making it easier for stochastic gradient descent to quickly find a good solution, limiting the novelty of this insight.
3) $\textit{“wide and shallow ones show […] smaller generalization gap”}$: this again is not directly observable from the empirical results. For instance, the widest and most shallow topology (width P = 4, depth d=0.5) has a higher gap between test and training loss than the most narrow and less shallow topology (width P = 1, depth d=1).

The same observations can be made for the experiments with CIFAR-10 and CIRFAR-100.

---

> ### Author Response · Authors · 2023-04-27
> **Response to Reviewer VWUy**
>
> ### 1. Depth and width of the ResNet block in Figure 2
> This ResNet block is interpreted as two parallel **paths** ($P_H$ = 2), instead of two layers. The skip-connection path has a depth of 0, while the other path has a depth of 3 (since there are three Linear-ReLU operations).
>
> ### 2. Total number of architectures
> When H = 3, there are 6 edges in the graph, with each edge being free to choose from 3 available operations (Linear-ReLU, Skip-connection, Zero). We thus have in total $3^6 = 729$ architectures. This is also true for our experiments: we trained all 729 architectures. Furthermore, we trained each network from this set three times with different randomness.
>
> ### 3. “comparison under the same training protocol may be unfair”
> We really appreciate this question!
>
> It is worth noting that current NAS benchmarks (e.g., NAS-Bench-101, NAS-Bench-201) employ fixed training hyperparameters across all networks, which may not be optimal. In this work, we have opted to follow the convention established by these benchmarks to ensure compatibility with the broader AutoML community.
>
> Meanwhile, we *are* currently undertaking further research to analyze the impact of architecture-aware training hyperparameters on network performance.
>
> ### 4. “networks of deep and narrow topology have higher expressivity”
> Our findings indicate that, given a fixed budget of the number of parameters, as illustrated in Figure 3 Left, the converged training loss is positively correlated ($R(P) = 0.11$) with the network’s topological width, indicating that wider topology will have higher training loss. Our training loss is small because we train networks for a long time (​​3000 epochs) to ensure that they are converged.
>
> ### 5. “wide and shallow ones show faster convergence”
> Most existing works focus on network’s channel width (i.e. number of neurons) of single layers, in the context of feedforward networks. Our work generalizes this result to complicated **graph topologies**, instead of the channel width of each layer.
>
> ### 6. “wide and shallow ones show […] smaller generalization gap”
> We verify this conclusion on three datasets: CIFAR-10, CIFAR-100 (Figure 5), and Tiny Imagenet (Figure 3). We calculate the correlation between the generalization gap vs. topological depth ($R(d_p)$) and width ($R(P)$). From the right columns of both Figure 3 and 5, we find a positive correlation with depth and a negative correlation with width. This confirms our conclusion that wide and shallow networks show smaller generalization gap.
>
> ### 7. Domain of image recognition only
> Our experiments are on image recognition mainly because most highly cited NAS benchmarks (NAS-Bench-101, NAS-Bench-201) are for image recognition, and we follow this convention to meet the trend of the broad AutoML community. Meanwhile, our theory does not assume any data modality.
>
> ### 8. Other operations
> Thanks for this question! We will try to expand our operation set to include convolution, attention, and other layers.
>
> It is worth noting that our experiments were rigorous despite using only three operations. We trained 729 architectures on three datasets, with each architecture trained for 3000 epochs and tested across three levels of randomness.
>
> Meanwhile, the implication of our work on the convolutional network is beyond Linear-ReLU/Skip/Zero operations, as listed in Section 3.2.5.
>
> ### 9. “skip-connection can improve the number of neurons from exponential to polynomial of the network depth.”
> This means skip-connection can improve (reduce) the requirement on the channel number from exponential to polynomial of the network depth. In other words, the convergence result will hold without requiring the network to be exponentially wide. We have revised our draft with a better explanation.
>
> ### 10. Better elaboration of Section 4.2
> Thank you! We followed your suggestion and have moved Lemma 4.1 to the appendix and added more explanation for Theorem 4.1 in red lines.

---

> > ### Comment · Reviewer_VWUy · 2023-05-03
> > **Response to Authors**
> >
> > Thank you for the response! While certain aspects are more comprehensible now and the overall clarity of the paper could be improved, the two most critical points remain unresolved:
> >
> > 1. Since you are defining the depth of a neural network as the average over all paths, the residual block has an average depth of (0 + 3) / 2 = 1.5, while the same architecture without the skip-connection has an average depth of 3 / 1 = 3. A justification for this has yet to be provided.
> >
> > 2. I understand that the architecture space of H = 3 includes 729 architectures. However, architectures with a width of 1 (single path) can only have operations of type "Linear-ReLU" or "Zero", since "Skip-connection"does not make sense for a single path. Thus, there is only a single architecture with width of 1 and depth of 3 (all nodes are connected with "Linear-ReLU") [o -> o -> o -> o], 4 architectures with a width of 1 and a depth of 2 (nodes that are connected with "Linear-ReLU": {0,1,2}, {0,1,3}, {0,2,3}, {1,2,3}) [o -> o -> o], and 6 architectures with a width of 1 and a depth of 1 (nodes that are connected with "Linear-ReLU": {0,1}, {0,2}, {0,3}, {1,2}, {1,3}, {2,3}) [o -> o]. This influences the expressivness of your results.
> >
> > Also, a correlation coefficient of 0.11 only suggests a weak positive correlation (point 4). Given that the hyperparameters have not been optimized for the different architectures (point 3), the implications of this results are limited and should be interpreted with caution.
> > Last but not least, point 7 is indeed well-founded, it was just not effectively conveyed in the main paper.
> >
> > In light of the provided information, I will maintain my current score.

---

> > > ### Author Response · Authors · 2023-05-09
> > > **Further Response to Reviewer VWUy**
> > >
> > > Dear reviewer VWUy,
> > >
> > > Thank you very much for your important questions!
> > >
> > > It took us some extra time to run more experiments and better prepare this response.
> > >
> > > 1. Yes, your point of view is correct, which is why in Figure 2 we wrote $d_0 = 3$ for MLP and $d_0 = 0, d_1 = 3$ for ResNet Block (i.e. an average depth of 1.5).
> > >
> > > 2. It is true that there is only one architecture with a width of 1 and a depth of 3. However, there are **25 architectures with a width of 1 and a depth of 2**. For each list below, we have an input node and three hidden nodes, and integers indicate operations connecting with previous nodes. "0" indicates "Zero" (broken edge), "1" indicates "Skip-connect", "2" indicates "Linear-ReLU". For example, "[[0], [2, 0], [0, 0, 2]]" means "Zero" b/w input and node 1; "Linear-ReLU" b/w input and node 2; "Zero" b/w node 1 and node 2; "Zero" b/w input and node 3; "Zero" b/w node 1 and node 3; "Linear-ReLU" b/w node 2 and node 3. Certainly, the "Zero" operator probably makes some architectures identical eventually, but they are still valid computational graphs in the architecture space, and they contribute to randomness of dots in our plots.
> > > * [[0], [2, 0], [0, 0, 2]]
> > > * [[0], [2, 0], [0, 1, 2]]
> > > * [[0], [2, 0], [0, 2, 2]]
> > > * [[0], [2, 1], [0, 0, 2]]
> > > * [[0], [2, 1], [0, 1, 2]]
> > > * [[0], [2, 1], [0, 2, 2]]
> > > * [[0], [2, 2], [0, 0, 2]]
> > > * [[0], [2, 2], [0, 1, 2]]
> > > * [[0], [2, 2], [0, 2, 2]]
> > > * [[1], [0, 2], [0, 0, 2]]
> > > * [[1], [2, 0], [0, 0, 2]]
> > > * [[2], [0, 0], [0, 2, 0]]
> > > * [[2], [0, 0], [0, 2, 1]]
> > > * [[2], [0, 0], [0, 2, 2]]
> > > * [[2], [0, 1], [0, 0, 2]]
> > > * [[2], [0, 1], [0, 2, 0]]
> > > * [[2], [0, 2], [0, 0, 1]]
> > > * [[2], [0, 2], [0, 2, 0]]
> > > * [[2], [1, 0], [0, 2, 0]]
> > > * [[2], [1, 1], [0, 2, 0]]
> > > * [[2], [1, 2], [0, 2, 0]]
> > > * [[2], [2, 0], [0, 0, 2]]
> > > * [[2], [2, 0], [0, 2, 0]]
> > > * [[2], [2, 1], [0, 2, 0]]
> > > * [[2], [2, 2], [0, 2, 0]]
> > >
> > > Similarly, there are 114 architectures with a width of 1 and a depth of 1.
> > >
> > > 3. We follow your recommendation, and prepare a comprehensive experiment with **tailored learning rates for each architecture**.
> > > We study the empirical expressivity, trainability, and generalization by training networks with their optimal learning rates tuned by grid search.
> > >
> > > We attach our revised draft in this anonymous repo: https://anonymous.4open.science/r/anonymous_AutoML2023_59-7058/AutoML2023_59.pdf
> > >
> > > As shown in Figure 3 in our Section C.3, even with architecture-wise optimal learning rates, our conclusion still holds: in a complete architecture space, given a fixed budget of network parameters, deep narrow networks have higher expressivity, while wide shallow networks converge faster and generalize better.
> > >
> > > We attach the optimal learning rates we find for each architecture in the anonymous repo ("lrs.txt"): https://anonymous.4open.science/r/anonymous_AutoML2023_59-7058/lrs.txt

---

> > > > ### Comment · Reviewer_VWUy · 2023-05-09
> > > > **Further Response to Authors**
> > > >
> > > > Thank you once again for the clarification.
> > > >
> > > > Concerning points 1 and 3, although you effectively eliminate one potential cause of misconception by using tailored learning rates, the current definition of 'depth' could still affect the conclusions drawn. For instance, the observed results could primarily be due to the presence or absence of skip connections rather than being attributed to the depth or shallowness of the network (correlation vs causation). An ablation study might provide valuable insights in this regard.
> > > >
> > > > Concerning point 2, while including "broken connections" indeed increases the number of evaluated architectures, the distribution remains uneven, ranging from a single (!) evaluated architecture to 114 evaluated architectures for a width of 1. As initially suggested, a more balanced approach to the distribution of evaluated architectures could enhance comparability.

---

> > > > > ### Author Response · Authors · 2023-05-09
> > > > > **Further Response to Reviewer VWUy**
> > > > >
> > > > > Dear reviewer VWUy,
> > > > >
> > > > > We appreciate your timely response and discussion!
> > > > >
> > > > > 1. Shu et al. [1] have a definition of depth that shares similar motivation with us, i.e., to characterize the graph topology of networks in NAS, and we also share similar architecture spaces (which include skip connections). Their observations are accepted by the AutoML community. Studying an architecture space with the skip connection is also common in the AutoML community [2].
> > > > >
> > > > > Meanwhile, would you suggest any concrete ablation study that could clear your concern?
> > > > >
> > > > > 2. We would like to point out:
> > > > >
> > > > > i) The motivation for projecting the high-dimensional architecture space to two principal directions (topological depth and width) is not to pursue a balanced distribution of architectures in this 2D space. Instead, it is because we find that the topological depth and width can well-distinguish the behavior of networks (w.r.t. expressivity/convergence/generalization). As an analogy, PCA never projects data to be a balanced distribution, but to preserve feature variances.
> > > > >
> > > > > ii) Again, numerous NAS papers work on "unbalanced" architecture spaces, including [1] and [2].
> > > > >
> > > > >
> > > > > [1] "Understanding Architectures Learnt by Cell-based Neural Architecture Search" Shu et al. 2019.
> > > > >
> > > > > [2] "NAS evaluation is frustratingly hard" Yang et al. 2020

---

> ### Author Response · Authors · 2023-05-01
> **Look forward to further discussions**
>
> Dear reviewer VWUy,
>
> Since the author-reviewer discussion period will end soon, we will highly appreciate it if you could check our updated draft and our response to your review comments at your earliest convenience. This way, if you have further questions and comments, we can still have time to reply.
>
> If our response resolves your concerns, we kindly ask you to consider raising the rating of our work.
>
> Thank you very much for your time and efforts!
>
> Sincerely,
>
> Authors of submission 59

---

### Official Review · Reviewer_7Zmj · 2023-04-11

**Potential Impact On The Field Of Automl Rating:** 3
**Technical Quality And Correctness Rating:** 4
**Clarity Rating:** 4

**Summary Of Contributions:**

The paper formalizes expressivity, convergence rate and generalization gap of a neural network based on its graph topology, and discover that given such representation and a fixed parameter budget, it is not possible to achieve optimal expressivity, convergence and generalization. This is because given the graph formulation of architectures, where Ph stands for number of end-to-end unique paths, and dp being the number of linear transformation operations on the p-th path, expressivity requies smaller Ph and larger dp, whereas convergence rate and generalization require larger Ph and lower dp. This is further empirically justified in a fixed setting in Figure 3 and Figure 4.

**Actions Required To Increase Overall Recommendation:**

Comments on whether their toy search space is sufficiently large/diverse for verifying their claims would help.

**Clarity:**

The paper is very clear, and all necessary proofs are provided in the Appendix. Contributions are clear and it is simple to understand the primary matter.

**Overall Review:**

This is a well written paper, with limited but effective empirical testing. Their formalization of expressivity, generalization and convergence is well justified and supported with conclusions verified by Figure 3 and Figure 4. The limitations are discussed in the conclusion, which would motivate future work.

I would like the authors views on including discussions regarding the implications of the structure of the linear transformations e.g. instead of an MLP, would convolutions/transformer based architectures and other aspects of the operations themselves (stride/pooling etc.) change the conclusions in the paper?

Discussion on whether their toy search space is sufficient to support their conclusions would also help.



**Potential Impact On The Field Of Automl:**

This paper jointly analyzes three key aspects of a neural network architecture, its expressivity, trainability and generalization. These are key aspects of development of NAS strategies, such as zero cost proxies for accuracy. I believe that the analysis provided by the authors will encourage further work in this field with a stronger focus on multiple aspects of NN design, aside from test accuracy.

**Review Confidence:**

2: You are willing to defend your assessment, but it is quite likely that you did not understand the central parts of the submission or that you are unfamiliar with some pieces of related work.

**Review Rating:**

7: Weak Accept: Technically sound paper with moderate-to-high impact and strong evaluation, with perhaps some minor flaws.

**Review Summary:**

I am somewhat confident that this paper should be accepted, it is well written with proofs and empirical evidence on a small search space of 729 neural networks to justify their observations derived from their formalization of expressivity, generalization and convergence. It is possible that I did not understand some parts of the submission (generalizability theorems) due to lack of familiarity with prior work.

**Technical Quality And Correctness:**

Expressivity

To study the expressivity, functional complexity of deep networks is studied. To aid this, a simple circle input is utilized to study the jacobian norm of an architecture. Theorem 4.1 provides insights on how to minimize the jacobian norm, with Corollary 4.2 introducing the curvature. Figure 3 verifies the relation between curvature and graph topology on a toy problem. I find the empirical conclusion consistent with observations from Eqn(4).

Convergence

Convergence is related to the networks least eigenvalue of its NNGP kernel. Theorem 4.4 bounds the least eigenvalue of NNGP kernel of the network, and maximize Ph followed by minimizing number of linear transformations on the edges. They remark that an upper bound does not meaningfully reflect its dependence on graph topology, and demonstrate that wider and shallowe networks correspond to higher \lambda_{min}(K^{H}). Is the MSE loss a limitation of the conclusions that follow in Section 4.3? A short discussion on that may help readers.


Generalization

Minimization of the leading term of the generalization bound Eqn (9) motivates their approach to maximize generalizability. This is further empirically verified by Figure 4. I was unable to access the validity of the theory due to lack of familiarity with relevant literature.

---

> ### Author Response · Authors · 2023-04-27
> **Response to Reviewer 7Zmj**
>
> ### 1. MSE loss in Theorem 4.4
>
> Although in Theorem 4.4 we used MSE loss, in our experiments we generalize this conclusion to the cross-entropy loss, which removes the possible limitation.
>
> ### 2. More layer types
>
> Thanks for this question! We will try to expand our operation set to include convolution, attention, and other layers.
>
> Meanwhile, as the implication of our work on the convolutional network is beyond Linear-ReLU/Skip/Zero operations (listed in Section 3.2.5), our core conclusion, the “No Free Lunch” behavior remains unchanged. The primary difference may lie in the requirements on the number of channels (for example, the requirement is multiplied by poly(p) from Theorem 6.1 to Theorem 7.1 in [1]).
>
> It is worth noting that our experiments were rigorous despite using only three operations. We trained 729 architectures on three datasets, with each architecture trained for 3000 epochs and tested across three levels of randomness.
>
> [1] “Gradient Descent Finds Global Minima of Deep Neural Networks” Du et al. 2018

---

> > ### Comment · Reviewer_7Zmj · 2023-05-02
> > **Response To Authors**
> >
> > Thank you for your response.
> >
> > I would like to keep my score and maintain my current confidence in my evaluation.
> > I strongly feel that a more in-depth empirical evaluation may be very valuable (in terms of the architecture design space). I will look at the discussions with the other reviewers and re-asses with higher confidence if possible once those discussions conclude.

---

### Official Review · Reviewer_YQ71 · 2023-04-12

**Potential Impact On The Field Of Automl Rating:** 2
**Technical Quality And Correctness Rating:** 3
**Clarity Rating:** 3

**Summary Of Contributions:**

The paper main contributions are as follows:
• Theoretical analysis of the dependence of a deep network’s manifold complexity, convergence rate, and generalization gap on its graph topology.
• Discovery of the “no free lunch” behaviour: given a fixed budget on the number of parameters pre-defined in an architecture space, no such a network can achieve optimal expressivity, convergence and generalization at the same time.
• Analysis explaining a wide range of observations in AutoML and NAS practices. Experiments on popular datasets confirm the theoretical analysis.

**Actions Required To Increase Overall Recommendation:**

To generliaze the results obtained from this study, different network should be studied. Additionally, different networks may have different optimal learning rate. Therefore, it may be deemed unfair to compare them under identical training protocols.


**Clarity:**

The paper is well-written and clearly explains complex technical concepts in a way that is accessible to a broad audience.

**Overall Review:**


Pros
1. sheds light on the limitations and challenges  of optimizing neural networks in an architecture space
2.  The paper is well-written and clearly explains complex technical concepts in a way that is accessible to a broad audience.
3. The paper demonstrates a comprehensive understanding of the technical literature in the field, and builds on it in a novel and innovative way
3.

Cons
1. To generalise the results obtained from this study, different network and datasets should be included.
2. different networks may have different optimal learning rate. Therefore, it may be deemed unfair to compare them under identical training protocols




**Potential Impact On The Field Of Automl:**


This work sheds light on the limitations and challenges  of optimizing neural networks in an architecture space. We challenge the prevailing assumption that there exists a single optimal network architecture and instead highlight the importance of balancing these three aspects when optimizing neural networks.  Additionally, this research is crucial for improving the performance of automated machine learning and neural architecture search algorithms, as we provide a theoretical foundation for designing better architectures.

**Review Confidence:**

3: You are fairly confident in your assessment. It is possible that you did not understand some parts of the submission or that you are unfamiliar with some pieces of related work.

**Review Rating:**

7: Weak Accept: Technically sound paper with moderate-to-high impact and strong evaluation, with perhaps some minor flaws.

**Review Summary:**


The paper facilitates the explanation of the architecture bias in AutoML and NAS applications through jointly analysing a network’s expressivity, trainability, and generalization, and how they are influenced by the architecture’s graph topology. Given a fixed budget of the number of parameters, it has been shown that the expressivity favors networks of deep and narrow graph topologies, whereas both
the trainability and generalization prefer wide and shallow ones. The paper is the first to discover that these inductive biases lead to a “no free lunch” behavior in deep network architectures. The paper is well-written and easy to follow.

However To generliaze the results obtained from this study, different network should be studied. Additionally, different networks may have different optimal learning rate. Therefore, it may be deemed unfair to compare them under identical training protocols.


**Technical Quality And Correctness:**

The paper demonstrates a high level of technical correctness and accuracy in its analysis and findings.

---

> ### Author Response · Authors · 2023-04-27
> **Response to Reviewer YQ71**
>
> ### 1. Different networks and datasets
> Thank you for your feedback. Our experiments were designed to focus on the graph-based architecture space for image recognition, primarily due to the fact that most highly cited NAS benchmarks (NAS-Bench-101, NAS-Bench-201) are graph-based and used for image recognition. We followed this convention to align with the broader AutoML community.
>
> Moving forward, we aim to include a wider range of architectures and datasets in our research.
>
> ### 2. Different networks may have different optimal learning rates.
> We really appreciate this question!
>
> It is worth noting that current NAS benchmarks (e.g., NAS-Bench-101, NAS-Bench-201) employ fixed training hyperparameters across all networks, which may not be optimal. In this work, we have opted to follow the convention established by these benchmarks to ensure compatibility with the broader AutoML community.
>
> Meanwhile, we *are* currently undertaking further research to analyze the impact of architecture-aware training hyperparameters on network performance.

---

### Official Review · Reviewer_73uc · 2023-04-13

**Potential Impact On The Field Of Automl Rating:** 2
**Technical Quality And Correctness Rating:** 4
**Clarity Rating:** 2
**Actions Required To Increase Overall Recommendation:** More explanations on the impact in th…

**Summary Of Contributions:**

This work investigates how the architecture of a neural network impacts three metrics: The “expressivity” meaning how well the next work can approximate the training data, the “convergence” meaning how fast the network learns, and the “generalization” meaning how well the network performs on unseen data.  Given a fixed budget of parameters, a “no free lunch” behavior can be observed, meaning that no model from a NAS space can achieve optimal performance in all three metrics, given a fixed amount of parameters. The authors derive the claims from a theoretical point of view and perform experimental evaluations on Nas-bench-102 to verify the claims.


**Clarity:**

Each paragraph of the paper is complete and clear in itself, however, the general structure of the paper lacks some clarity. The authors first conclude their main contribution (the discovery of a “no free lunch” behavior) already on page 5 in Section 3, while then from pages 6-9 showing the theoretical derivations for their claims. This makes it hard to relate the three paragraphs “Expressivity/Convergence/Generalization Analysis of Architectures” to the main claims in Section 3.


**Overall Review:**

Positives:

- The results are interesting from a deep learning theory perspective.
- The formal results section as well as the experiments are extensive.
- The three metrics are an interesting way to characterize the performance of a neural network.

Negatives:

- The authors claim in the abstract that the default measure of performance in AutoML (the test accuracy) is closely related to the three aspects (expressivity/convergence/generalization). However, this is only explained for the aspect of ‘generalization’ in Section 4.4, but not for the concepts of expressivity and convergence. As far as I can see, this is a major flaw, as it limits the conclusion that insights can be derived for the AutoML community.
- The “Architecture bias in AutoML” section (lines 192-196) is shallow, given that here the “no free lunch” behavior is put in context to applications in AutoML. A more detailed explanation is necessary.

**Potential Impact On The Field Of Automl:**

The analysis provided in the paper is interesting from a theory of deep learning perspective, but since the goal of AutoML is to find top-performing architectures it is not clear to me why the number of total parameters is a fixed value in the NAS setting.


**Review Confidence:**

2: You are willing to defend your assessment, but it is quite likely that you did not understand the central parts of the submission or that you are unfamiliar with some pieces of related work.

**Review Rating:**

5: Borderline Leaning Reject: Technically sound paper where reasons to reject nonetheless outweigh reasons to accept. Please use sparingly.

**Review Summary:**

Overall, the paper presents interesting results and supports them with theoretical derivations and particle experiments. My main concern is that the impact on the field of AutoML is limited.


**Technical Quality And Correctness:**

The theoretical derivations presented in the main paper are exhaustive, as well as the experimental evaluations on several datasets in the main paper and the appendix.

---

> ### Author Response · Authors · 2023-04-27
> **Response to Reviewer 73uc**
>
> ### 1. Paper Structure
> Thank you for your valuable feedback! We have taken your concerns into consideration and have made revisions to our draft by establishing more connections between Section 3 and Section 4, as well as including red lines and subtitles to better illustrate these connections.
>
> We have chosen to defer formal results to Section 4 in order to present our core findings and implications to wider AutoML audiences without being overwhelmed with theoretical details.
>
> We are open to further revisions based on your feedback! As confirmed by reviewer YQ71, our draft is “well-written and clearly explains complex technical concepts in a way that is accessible to a broad audience” and by reviewer 7Zmj that our draft is “very clear.”
>
> ### 2. Explanation: why test accuracy is closely related to three aspects (expressivity/convergence/generalization)
> We appreciate your question. It is important to note that the (decomposition of) test loss is directly linked to expressivity, convergence, and generalization.
>
> This is because: related generalization bounds [1, 2], including ours (Theorm 4.5), are based on the assumption that wide networks can achieve small training errors. See Section 4 in [1] “predicts linear convergence of GD to 0 loss” and Remark 3.4 in [2] “Theorem 3.3 suggests that if the data can be classiﬁed … with a small training error, the over-parameterized ReLU network learned by Algorithm 1 will have a small generalization error.”
>
> Therefore, it is crucial to ensure that the network can achieve a low training error (i.e., high expressivity), and we hope to achieve this with fewer steps (i.e. faster convergence). That means, the training error and the convergence rate are not explicitly characterized in Theorem 4.5, which makes our analysis of expressivity and convergence necessary.
>
> We will make sure to include this explanation in our camera ready.
>
>
> ### 3. “Architecture bias in AutoML” is shallow
> Thank you for your input. We have revised this point to provide more precise context, focusing specifically on differentiable NAS, and have included additional explanations.
>
> [1] “Fine-Grained Analysis of Optimization and Generalization for Overparameterized Two-Layer Neural Networks” Arora et al. 2019
>
> [2] “Generalization Bounds of Stochastic Gradient Descent for Wide and Deep Neural Networks” Cao et al. 2019

---

> > ### Comment · Reviewer_73uc · 2023-05-01
> > **Response to Authors**
> >
> > I thank the authors for their response.
> >
> > 1. Regarding the paper structure, I acknowledge that the current structure was chosen to present the results in the first part of the manuscript and defer the theoretical explanations to the second part which is a valid reason.
> >
> > 2. I also appreciate the explanations on the decomposition of test accuracy metric.
> >
> > 3. However, I still find the section on the "Architecture bias in (differentiable) NAS" to not provide enough context to explain the impact of the findings on applications in AutoML. As the focus of this venue is specifically on AutoML, I see this section as one of the most important parts of the paper that would warrant an in-depth explanation (which should also be feasible given that there is still roughly one more page left to the page limit).

---

> > > ### Author Response · Authors · 2023-05-01
> > > **Thanks for your feedback!**
> > >
> > > Dear reviewer 73uc,
> > >
> > > We are deeply grateful for your prompt and insightful feedback on our manuscript.
> > >
> > > In response to your suggestions, we have revised and expanded Section 3.2.5 to provide a more comprehensive context and clearer explanations of our impact and findings regarding AutoML. We acknowledge the significance of this section for the AutoML community and have made the necessary improvements.
> > >
> > > We kindly request that you review the updated Section 3.2.5. We are committed to making additional revisions if needed to ensure the highest quality of our work.
> > >
> > > Sincerely,
> > > Authors of submission 59

---

> > > > ### Comment · Reviewer_73uc · 2023-05-02
> > > > **Explanation on AutoML Impact**
> > > >
> > > > I thank the authors for extending the section. I remain skeptical, especially as the explanation in line 213 "Therefore, the differentiable search tends to favor networks that can minimize training loss as quickly as possible." sounds more like an overfitting problem to me, which would be obvious in a network with not enough learnable parameters. I am therefore raising my score only from weak reject to borderline.

---

> > > > > ### Author Response · Authors · 2023-05-02
> > > > > **Thanks for your feedback!**
> > > > >
> > > > > Dear reviewer 73uc,
> > > > >
> > > > > We appreciate your continued engagement in the discussion!
> > > > >
> > > > > We would like to elaborate on the following points:
> > > > > 1. This collapse issue happens at the early stage of DARTS optimization (that is why "early stopping" is proposed to mitigate this problem [1]).
> > > > > At this stage, network parameters have not been fully trained and the training loss remains high. Consequently, the DARTS supernet and its subnetworks are not in an overfitting state.
> > > > > 2. "networks with not enough learnable parameters" usually tend to underfit, not overfit.
> > > > >
> > > > > We hope this clarification addresses your concerns, and we are happy to further discuss before the rebuttal period ends.
> > > > >
> > > > > Sincerely,
> > > > >
> > > > > Authors of submission 59
> > > > >
> > > > > [1] "DARTS+: Improved Differentiable Architecture Search with Early Stopping" Liang et al. 2020

---

> > > > > ### Author Response · Authors · 2023-05-09
> > > > > **Look forward to any further feedback and discussions**
> > > > >
> > > > > Dear reviewer 73uc,
> > > > >
> > > > > Per the AutoML official policy, we can still read and respond to reviews/reviewer comments. Thus, we would highly appreciate it if you could check our latest response to your comment. This way, if you have further questions or comments, we can still reply in a timely manner.
> > > > >
> > > > > If our response resolves your concerns, we kindly ask you to consider raising the rating of our work.
> > > > >
> > > > > Thank you very much for your time and efforts!
> > > > >
> > > > > Sincerely,
> > > > >
> > > > > Authors of submission 59

---

> ### Author Response · Authors · 2023-05-01
> **Look forward to further discussions**
>
> Dear reviewer 73uc,
>
> Since the author-reviewer discussion period will end soon, we will highly appreciate it if you could check our updated draft and our response to your review comments at your earliest convenience. This way, if you have further questions and comments, we can still have time to reply.
>
> If our response resolves your concerns, we kindly ask you to consider raising the rating of our work.
>
> Thank you very much for your time and efforts!
>
> Sincerely,
>
> Authors of submission 59

---

### Review · Reproducibility_Reviewer_D5bQ · 2023-04-13

**Completeness Of Code And Dataset Supplement Rating:** 3
**Usability And Ease Of Reproducibility Rating:** 3

**Actions Required To Increase The Reproducibility And Overall Recommendation:**

Please provide more detailed instructions for setting up and running your code, as well as explanations in your checklist.

**Completeness Of Code And Dataset Supplement:**

The code supplement does contain the core code required to replicate their experiment. The visualization code does not appear to be included.

**Overall Reproducibility Review:**

Overall, the code and information provided in the supplement is sufficient for replicating the described work. However, it is not provided with clear and sufficient instructions on how to set up and use the code. The absence explanations in the checklist does not assist with this lack of clarity.

**Review Confidence:**

4: You are confident in your assessment, but not absolutely certain. It is unlikely, but not impossible, that you did not understand some parts of the submission or that you are unfamiliar with some pieces of the code or data.

**Review Rating:**

8: Accept, all aspects of this are reproducible with minor effort.

**Review Summary:**

n/a

**Summary Of Necessary Code And Dataset Supplement:**

The code required for this includes the code to train each candidate neural network in their search space, as well as the code to evaluate the expressivity, convergence and generalization of each of those architectures. Additionally, plotting each of these metrics requires visualization code. Data sets used include cifar10, cifar100 and tiny-imagenet.

**Usability And Ease Of Reproducibility:**

The code has some significant issues with usability. There is no included "requirements.txt" or similar instructions on how to install the prerequisite software, which makes it time consuming to set up a proper environment for running the code. Similarly, there are not instructions for downloading or details about sett up the necessary data sets for running the code. The provided instructions don't make a clear distinctions between running the computational portion of the main text and the ensembling experiment whose results are detailed in the supplement, which makes the use of the code somewhat more confusing. The file headers for each experiment-launching python file are all the same, which does not help to disambiguate. The code here is feasible to run, it isn't unusable by any means, but it is not packaged for easy replication.

---

> ### Author Response · Authors · 2023-04-27
> **Response to Reviewer D5bQ**
>
> Thank you very much for these great suggestions! We have uploaded our code supplement:
> * We have included the “requirements.txt”
> * We have included more instructions (e.g. datasets, installation) in our README.md
> * We have the command for running network ensembles under “Prune DAG Ensemble” in README.md
> * We have updated file headers.
> * We have included the jupyter notebook for visualizations.
>
> We are happy to discuss more on improving our reproducibility!

---

### Author Response · Authors · 2023-04-27
**Rebuttal by Authors**

We truly thank all questions and suggestions from the three reviewers. We are happy that reviewers YQ71 and 7Zmj acknowledge that we work on an important problem for the AutoML community, with well-conducted experiments, and also acknowledge our clear paper writing. We also appreciate reviewer D5bQ’s confirmation of our reproducibility.

We address all questions below.

---

### Author Response · Authors · 2023-04-28
**Look forward to any further feedback and discussions**

Dear all reviewers,

Since the author-reviewer discussion period has started for a few days, we will appreciate it if you could check our updated draft and our response to your review comments soon. This way, if you have further questions and comments, we can still reply before the author-reviewer discussion period ends.

If our response resolves your concerns, we kindly ask you to consider raising the rating of our work.

Thank you very much for your time and efforts!

Sincerely,

Authors of submission 59